

# Long-range transport pathways of tropospheric source gases originating in Asia into the northern lower stratosphere during the Asian monsoon season 2012

Bärbel Vogel[1], Gebhard Günther[1], Rolf Müller[1], Jens-Uwe Grooß[1], Armin Afchine[1], Heiko Bozem[2], Peter Hoor[2], Martina Krämer[1], Stefan Müller[2], Martin Riese[1], Christian Rolf[1], Nicole Spelten[1], Gabriele P. Stiller[3], Jörn Ungermann[1], and Andreas Zahn[3]

[1]Forschungszentrum Jülich, Institute of Energy and Climate Research - Stratosphere (IEK-7), Jülich, Germany
[2]Institute for Atmospheric Physics, University of Mainz, Mainz, Germany
[3]Institute for Meteorology and Climate Research, Karlsruhe Institute of Technology, Karlsruhe, Germany

*Correspondence to:* Bärbel Vogel
(b.vogel@fz-juelich.de)

**Abstract.** Global simulations with the Chemical Lagrangian Model of the Stratosphere (CLaMS) using artificial tracers of air mass origin are used to analyze transport pathways from the Asian monsoon region into the lower stratosphere. In a case study, the transport of air masses from the Asian monsoon anticyclone originating in India/China by an eastward migrating anticyclone breaking off from the main anticyclone on 20 September 2012 and filaments separated at the northeastern flank of the anticyclone are analyzed. Enhanced contributions of young air masses (younger than 5 months) are found within the separated anticyclone confined at the top by the thermal tropopause. Further, these air masses are confined by the anticyclonic circulation and at the polar side by the subtropical jet such as the vertical structure looks like a bubble within the upper troposphere. Subsequently, these air masses are transported eastwards along the subtropical jet and enter the lower stratosphere by quasi-horizontal transport in a region of double tropopauses most likely associated with Rossby wave breaking events. As a result, thin filaments with enhanced signatures of tropospheric trace gases are measured in the lower stratosphere over Europe during the TACTS/ESMVal campaign in September 2012 in very good agreement with CLaMS simulations. Our simulations demonstrate that source regions in Asia and in the Pacific Ocean have a significant impact on the chemical composition of the lower stratosphere of the Northern Hemisphere by flooding the extratropical lower stratosphere with young moist air masses in particular at end of the monsoon season in September/October 2012 (up to $\approx 30\%$ at 380 K) in contrast to the southern hemisphere. End of October 2012, approximately 1.5 ppmv $H_2O$ is found in the lower northern hemisphere stratosphere (at 380 K) from source regions in Asia and the tropical Pacific compared to a mean water vapor content of $\approx 5$ ppmv. In addition to this main transport pathway from the Asian monsoon anticyclone to the east along the subtropical jet and subsequent transport into the northern lower stratosphere, a second horizontal transport pathway out of the anticyclone to the west into the tropics (TTL) is found in agreement with MIPAS HCFC-22 measurements.





## 1 Introduction

The Asian summer monsoon is associated with strong upward transport of tropospheric source gases by deep convection. Uplifted tropospheric air masses are confined by a strong anticyclonic circulation in the upper troposphere which is acting as a transport barrier (e. g. Li et al., 2005; Randel and Park, 2006; Park et al., 2007, 2008, 2009; Ploeger et al., 2015). In summer
2012, enhanced tropospheric trace gases are found in the upper troposphere over Asia from mid-June until late October both during the existence of the Asian monsoon anticyclone and after its breakup (Vogel et al., 2015). Studies for the summer 2012 show that air masses from the Asian monsoon impact the chemical composition of the lower stratosphere over northern Europe, in particular at the end of the monsoon season in September and October 2012 (Vogel et al., 2014, 2015; Müller et al., 2015). An increase in the concentrations of tropospheric source gases such as CO, $H_2O$, and $N_2O$ and a decrease of the stratospheric
tracer $O_3$ was identified in in-situ aircraft measurements in the lower stratosphere over Europe from summer to autumn 2012 associated with transport from the Asian monsoon anticyclone (Müller et al., 2015).

The identification and the relative importance of different transport pathways of tropospheric source gases found within the Asian monsoon anticyclone into the lower stratosphere are subject of a longstanding debate (Dethof et al., 1999; Park et al., 2004; Randel et al., 2010; Bian et al., 2012; Bourassa et al., 2012; Ploeger et al., 2013; Fairlie et al., 2014; Fromm
et al., 2014; Uma et al., 2014; Vogel et al., 2014, 2015; Garny and Randel, 2016; Tissier and Legras, 2016; Orbe et al., 2015). The influence of these different transport pathways on the chemical composition of the extratropical upper troposphere and lower stratosphere (ExUTLS) is important because changes in ozone and water vapor in the ExUTLS have a significant impact on surface climate, even if the perturbation is relatively small. Thus changes in the chemical composition in that part of the Earth's atmosphere play a crucial role for climate change (e. g. Solomon et al., 2010; Riese et al., 2012; Hossaini et al., 2015).
Moreover, increasing stratospheric water vapor has the potential to affect stratospheric chemistry (e. g. Kirk-Davidoff et al., 1999; Dvortsov and Solomon, 2001; Vogel et al., 2011a; Anderson et al., 2012) and can influence the formation of cirrus clouds in the lower stratosphere (Spang et al., 2015). Further, enhanced water vapor in combination with pollution in the Asian monsoon region is discussed to play an important role in the formation of the Asian Tropopause Aerosol Layer (ATAL) which causes a regional radiative forcing (Vernier et al., 2015).

One important mechanism for long-range transport of air masses from the Asian monsoon anticyclone to the extratropical lower stratosphere is the separation of air masses at the northeastern flank of the anticyclone caused by disturbances of the subtropical jet by strong Rossby waves and subsequent eastward transport of these air masses within the tropics along the subtropical jet (Dethof et al., 1999; Vogel et al., 2014, 2015). In particular, eastward migrating anticyclones breaking off from the main anticyclone a few times each summer (Dethof et al., 1999; Hsu and Plumb, 2001; Popovic and Plumb, 2001;
Garny and Randel, 2013; Vogel et al., 2014; Ungermann et al., 2016) have the potential to carry air with high amounts of tropospheric trace gases from the Asian monsoon anticyclone to mid- and high latitudes of the Northern Hemisphere. These air masses can be transported into the extratropical lower stratosphere where they are eventually mixed irreversibly with the surrounding stratospheric air (Dethof et al., 1999; Garny and Randel, 2013; Vogel et al., 2014) thus affecting the chemical and radiative balance of the extratropical lower stratosphere. However, the exact transport mechanisms of young air masses





from the troposphere including air originating within the Asian monsoon anticyclone into the lower stratosphere are yet to be identified and quantified.

Trajectory calculations show that tropospheric air masses at the southeast edge of the anticyclonic circulation over Southeast Asia may be uplifted to the outer anticyclonic circulation by a typhoon to levels of potential temperature of ≈370 K (Vogel et al., 2014). Subsequent upward transport of these air masses occurs in a clockwise upward spiral around the core of the Asian monsoon anticyclone to levels of potential temperature around 380 K. Moreover, Vogel et al. (2014) demonstrate that the combination of very rapid uplift by a typhoon and eastward eddy shedding from the Asian monsoon anticyclone is an additional rapid transport pathway (≈ 5 weeks) connecting surface air with enhanced signatures of tropospheric trace gases measured in the lower stratosphere over northern Europe.

Here, we use in-situ measurements obtained during two aircraft campaigns TACTS and ESMVal, jointly conducted in August and September 2012, using the German High Altitude and LOng Range Research Aircraft (HALO). Backward trajectory calculations (Vogel et al., 2014; Müller et al., 2015) show that enhanced tropospheric trace gases that were measured over northern Europe in the extratropical lower stratosphere during the TACTS/ESMVal campaign during August and September 2012 are affected by air masses from the circulation of the Asian monsoon anticyclone or originating in the Asian monsoon anticyclone itself.

Further model simulations with the Chemical Lagrangian model of the stratosphere (Vogel et al., 2015) using artificial emission tracers for different regions on the Earth's surface demonstrate that emissions of young air masses primarily from India, China, and Southeast Asia influence the chemical composition of the lower stratosphere of the Northern Hemisphere, in particular at the end of the monsoon season in September/October 2012. Thus the Asian monsoon anticyclone and boundary emissions from Asia contribute to the maximum of tropospheric signatures found in the northern extratropical lower stratosphere in boreal summer and autumn (e. g. Hoor et al., 2005; Hegglin and Shepherd, 2007; Bönisch et al., 2009; Zahn et al., 2014; Müller et al., 2015), sometimes also referred to as 'flushing' of the lower stratosphere (Hegglin and Shepherd, 2007; Bönisch et al., 2009).

In this paper, we identify in detail the transport mechanisms of air masses originating in boundary source regions in Asia and separated from the Asian monsoon anticyclone by eastward eddy shedding into the lower stratosphere over northern Europe including mixing processes. The same model simulation as in Vogel et al. (2015) with the three-dimensional Lagrangian chemistry transport model (CLaMS) (e. g. Pommrich et al., 2014, and references therein) are used including artificial emission tracers that mark source regions in the boundary layer of the Earth's atmosphere. This allows the origin of air masses and the detailed transport pathways from Asian source regions into the northern lower stratosphere to be quantified. Further, the irreversible part of transport, i .e. mixing, in CLaMS is controlled by the local horizontal strain and vertical shear rates with mixing parameters deduced from observations (Konopka et al., 2012, and references therein) and therefore allows mixing processes in the lower stratosphere as shown in several previous studies to be characterized (e. g. Pan et al., 2006; Vogel et al., 2011b; Konopka and Pan, 2012). Small scale structures indicating the impact of young tropospheric air masses within the lower stratosphere measured during the TACTS/ESMVal aircraft campaign in September 2012 are compared with results of CLaMS



model simulations to identify their origin and the associated transport pathways. Finally, the impact of boundary source regions in Asia on the composition of the extratropical Northern Hemisphere is calculated.

The paper is organized as follows: Section 2 describes the in situ measurements and Section 3 the setup for used CLaMS simulations. In Section 4, CLaMS results are presented and compared to measurements. A short discussion and conclusions are given in Sections 5 and 6.

## 2 TACTS/ESMVal Measurements over Europe

Here we use in-situ measurements performed during two aircraft campaigns TACTS and ESMVal, jointly conducted in August and September 2012. TACTS was designed to study 'Transport and Composition in the Upper Troposphere and Lowermost Stratosphere'. The objective of the ESMVal (Earth System Model Validation) measurement campaign was to validate model simulations with measurements ranging from the Southern to the Northern Hemisphere (here from 65°S to 80°N). Both campaigns were performed using the German High Altitude and LOng Range Research Aircraft (HALO), a Gulfstream V. During the TACTS/ESMVal campaign 13 research flights were performed, the most of them over northern Europe. Here, we discuss the last three flights conducted on 23, 25, and 26 September 2012. Measurements of the following in-situ instruments on board the HALO aircraft are used:

– Carbon monoxide (CO) and methane ($CH_4$) measurements are used from TRIHOP a three channel Quantum Cascade Laser Infrared Absorption spectrometer (for more details see Müller et al., 2015) which is an updated version of the three-channel tunable diode laser instrument for atmospheric research (TRISTAR) used in previous aircraft campaigns (e.g. Hoor et al., 2004).

– Water vapor ($H_2O$) measurements were obtained from the Fast In-situ Stratospheric Hygrometer (FISH) which is based on the Lyman-$\alpha$ photofragment fluorescence technique (Zöger et al., 1999). The FISH inlet was mounted forward-facing to measure total water which is the sum of gas-phase water and water in ice particles. To calculate gas-phase water from FISH measurements in clouds a correction procedure is applied (for more details see Meyer et al., 2015).

– Ozone ($O_3$) was measured with FAIRO a light-weight (14.5 kg) instrument with high accuracy (2 %) and high time resolution (10 Hz) developed for the HALO aircraft. FAIRO combines dual-beam UV photometer with an UV-LED as light source and a dry chemiluminescence detector (Zahn et al., 2012).

– Potential temperature ($\Theta$) was deduced from the Basic HALO Measurement and Sensor System (BAHAMAS) that yields basic meteorological and avionic data of all TACTS/ESMVal flights.




## 3 CLaMS simulations using artificial tracers of air mass origin

We use the same model simulation with the Chemical Lagrangian model of the stratosphere (McKenna et al., 2002b, a; Pommrich et al., 2014, and references therein) as in Vogel et al. (2015) covering the Asian monsoon season 2012. The CLaMS model has the capability to reproduce strong gradients of chemical species found in regions with strong transport barriers such as the

edge of the Asian monsoon anticyclone (e.g. Konopka et al., 2010; Ploeger et al., 2015; Vogel et al., 2015), the extratropical tropopause (e.g. Pan et al., 2006; Vogel et al., 2011b; Konopka and Pan, 2012) and the polar vortex (e.g. Günther et al., 2008; Vogel et al., 2008).

In brief, the global CLaMS simulation employed here covers an altitude range from the surface up to $900\,\mathrm{K}$ potential temperature ($\approx 37\,\mathrm{km}$ altitude) with a horizontal resolution of $100\,\mathrm{km}$ and a maximum vertical resolution of approximately

$400\,\mathrm{m}$ near the tropopause. The model simulation is driven by horizontal winds from ERA-Interim reanalysis (Dee et al., 2011) provided by the European Centre for Medium-Range Weather Forecasts (ECMWF). For the vertical velocities, the diabatic approach was applied using the diabatic heating rate as vertical velocity including latent heat release (for more details see Ploeger et al., 2010). Further, CLaMS employs a hybrid coordinate ($\zeta$), which transforms from a strictly isentropic coordinate $\Theta$ to a pressure-based coordinate system below a certain reference level (in this paper $300\,\mathrm{hPa}$) (for more details see Konopka

et al., 2012; Pommrich et al., 2014).

The CLaMS simulation covers the time period from 1 May 2012 until 31 October 2012 including the Asian monsoon season 2012. It includes full stratospheric chemistry (Grooß et al., 2014; Sander et al., 2011). Chemical species are initialized on 1 May 2012 mainly based on satellite data from AURA-MLS (version 3.3) (Livesey et al., 2011) and ACE-FTS (version 3.0) (Waymark et al., 2013) measurements. At the upper boundary ($900\,\mathrm{K}$ potential temperature), mainly AURA-MLS and

ACE-FTS measurements and tracer–tracer correlations were used (for more details see Vogel et al., 2015).

At the lower boundary (surface), $O_3$ is set to a constant tropospheric volume mixing ratio of $4.8 \times 10^{-8}$ volume mixing ratio representing the ozone mixing ratio at $5\,\mathrm{km}$ (Brasseur and Solomon, 2005, p. 619). ECMWF water vapor is prescribed at lower model levels. Lower boundary conditions for CO and $CH_4$ are derived from AIRS (Atmospheric Infrared Sounder) version 6 satellite measurements following the approach described by Pommrich et al. (2014).

In the CLaMS simulation used here, artificial tracers of air mass origin, referred to as "emission tracers", are implemented that mark globally all regions in the Earth's boundary layer ($\approx 2$–$3\,\mathrm{km}$ above the surface following orography corresponding to $\zeta < 120\,\mathrm{K}$) as shown in Fig. 1 and Tab. 1. Air masses in the model boundary layer are marked by different emission tracers every $24\,\mathrm{h}$ (the time step for mixing in CLaMS). Transport and mixing of the emission tracers to other regions of the free

troposphere or stratosphere occurs like for all chemical species included in the CLaMS simulation. The percentage of an individual emission tracer counts the contribution of the corresponding boundary layer region to the composition of an air parcel considering advection and mixing since 1 May 2012.

Emission tracers in CLaMS are designed to identify surface regions of the Earth's atmosphere that contribute to the composition of the Asian monsoon anticyclone and of the lower stratosphere during the course of the 2012 Asian monsoon season





with a focus on the influence of fresh emissions. Therefore, the starting point for the CLaMS simulation is chosen a few weeks before the formation of the Asian monsoon anticyclone occurred. Thus, transport processes associated with the Asian monsoon anticyclone and horizontal transport from the TTL into the lower stratosphere are considered.

Vogel et al. (2015) show that at the end of the monsoon season in September up to 75 % of the air masses within the Asian monsoon anticyclone are younger than 5 months. In addition, they show that the emission tracer for India/China (Northern India + Southern India + Eastern China) is a good proxy for the location and shape of Asian monsoon anticyclone using pattern correlations with potential vorticity (PV), and MLS $O_3$ and $CO$ measurements. Therefore, the emission tracer for India/China is very well suited to analyze transport pathways from the Asian monsoon anticyclone into the northern lower stratosphere. The emission tracer for Southeast Asia contributes also to the composition of the Asian monsoon but is in addition found in air masses circulating around the outer edge of the Asian monsoon anticyclone at levels of potential temperature of around 380 K.

## 4 Results

### 4.1 Eastward eddy shedding and separation of filaments

We will analyze transport mechanisms and pathways of tropospheric air masses separated by eastward eddy shedding or filaments at the northeastern flank of the Asian monsoon anticyclone into the lower stratosphere over Europe for the 2012 monsoon season.

At the northeastern flank of the Asian monsoon anticyclone filaments with low PV and enhanced percentages of the emission tracer for India/China are frequently separated from the main anticyclone in 2012, e.g. on 4 and 5 September 2012 as shown in Fig. 2. In addition, eastward migrating anticyclones break off from the main anticyclone several times during summer. A pronounced eddy shedding event occurred on 20 September 2012 (Vogel et al., 2014). A second anticyclone (130°E-210°E) characterized by low PV separated from the main Asian monsoon anticyclone (10°E-110°E) on 20 September 2012 as shown in Fig. 3 (top left). Thereafter, transport of these air masses characterized by low PV to the Pacific Ocean occurred (see Fig. 3, left) at 380 K.

Fig. 3 (right column) shows that air masses with enhanced contributions of the emission tracers for India/China and originating from the Asian monsoon anticyclone are transported eastwards within the separated anticyclone. After 20 September 2012, a long filament of air characterized by low PV and enhanced percentages of emission tracers for India/China evolves from the separated anticyclone. On 23 September 2012, this filament is located over North America along a poleward excursion of the subtropical westerly jet. Between 24 and 26 September 2012, these air masses are transported further eastwards over North America and afterwards to the Atlantic Ocean.

A very similar horizontal distribution as for the emission tracer for India/China is also found for simulated CO (see Fig. 4, left). However, enhanced CO values are in addition found at 380 K in the tropics associated with deep uplift in the tropics outside of the monsoon region to the upper troposphere in particular over the Maritime Continent and the western Pacific (see emission tracer for Southeast Asia and tropical Pacific Ocean in Appendix A to this paper). The emission tracer for Southeast Asia / tropical Pacific Ocean contributes also to the composition of the Asian monsoon anticyclone, however to a smaller extent



compared to the emission tracer for India/China as shown in Fig. 4 (right). Maximum percentages for the emission tracer for Southeast Asia / tropical Pacific Ocean are found at the edge of the Asian monsoon anticyclone indicating the transport of these air masses around the outer edge of the Asian monsoon anticyclone outside of a PV-based transport barrier (Ploeger et al., 2015) as discussed by Vogel et al. (2015). Thus air masses from Southeast Asia / tropical Pacific Ocean are found within
a widespread area around the anticyclone caused by the large-scale anticyclonic flow in this region acting as a large-scale stirrer. Moreover, high contributions of these emission tracers are found in the tropics associated with deep uplift out side of the monsoon region.

Kunz et al. (2015) derived a climatology of PV streamers (e.g. polward moving filaments with low PV originating in the tropics) as indicators of Rossby wave breaking on isentropes between 320 and 500 K using ERA-Interim reanalyses for the
time period from 1979 to 2011. The 7.2 PVU isoline shown in black in Fig. 3 and Fig. 4 represents the isentropic transport barrier between the tropics and mid-latitudes based on the climatology of the dynamically relevant PV contours derived by Kunz et al. (2015) at 380 K for the Northern Hemisphere for September until November. Isentropic transport of air masses across the 7.2 PVU isoline indicates irreversible exchange between the tropics and extratropics due to wave breaking. On 20 September 2012, at the northern flank of the separated anticyclone the 7.2 PVU isoline envelops the region of enhanced tracers
indicating the transport barrier at 380 K (see Fig. 3 and 4).

Considering altitude-height cross-sections from the separated anticyclone caused by eddy shedding at 180°E (Fig. 5) shows that air masses within the separated anticyclone (20°N – 40°N) throughout the troposphere are characterized by high percentages of the emission tracers for India/China (top left). The vertical tracer distribution of the separated anticyclone looks like a bubble that is confined by the subtropical westerly jet in the north (35°N– 55°N) and by the thermal tropopause at the top. The
thermal tropopause is slightly elevated similar as for the Asian monsoon anticyclone itself. Here, the thermal tropopause acts as a transport barrier for further upward transport from the separated anticyclone into the lower stratosphere. The terminology 'tropospheric bubble' is also used to describe the trace gas distribution within the Asian monsoon itself (Pan et al., 2016).

In addition, the isentropic transport barriers of 4 and 10 PVU at 350 K and 400 K (Kunz et al., 2015), respectively, are shown in Fig. 5 (white thick lines) for the Northern Hemisphere for September until November. No significant transport across
these PV isolines at the polar edge of the separated anticyclone was found indicating that here no air mass exchange between stratosphere and troposphere occurred due to Rossby wave breaking.

The altitude-height cross-section for the emission tracer for Southeast Asia / tropical Pacific Ocean (Fig. 5, top right) has also the structure of a bubble with enhanced percentages. These enhanced contributions of the emission tracer for Southeast Asia / tropical Pacific Ocean are clearly separated from the tropics, outside of the monsoon circulation, which shows even
higher contributions (up to 40%) of these tracers.

Fig. 5 (bottom left) shows the emission tracer for the residual surface that is the sum of all other emission tracers (all tracers without India/China and Southeast Asia / tropical Pacific Ocean). The emission tracer for the residual surface shows no signature within the separated anticyclone indicating that air masses originating in India/China and Southeast Asia / tropical Pacific Ocean almost exclusively contribute to the chemical composition of the separated anticyclone.





In general, the troposphere is characterized by weak static stability (buoyancy frequency squared, $N^2$) in contrast to stratospheric air masses. The altitude-height cross-section of $N^2$ (Fig. 5, bottom right) shows that air masses within the separated anticyclone are characterized by tropospheric $N^2$ values. Our model simulation demonstrates that air masses enclosed in the second anticyclone are still located in the troposphere. The question arises where exactly air masses from the Asian monsoon anticyclone separated by eddy shedding or large filaments will enter the lower stratosphere i. e. where they exactly cross the extratropical tropopause.

## 4.2 Isentropic transport pathways into the lower stratosphere

As shown in Figs. 3 and 4 (middle) a long filament of air characterized by low PV and high percentages of emission tracers for India/China and Southeast Asia / tropical Pacific Ocean evolves from the separated anticyclone. On 23 September 2012, this filament is located over North America along the subtropical westerly jet. At 380 K enhanced contributions of the emission tracer from India/China are also found north of the 7.2 PVU barrier indicating transport from the troposphere into stratosphere.

An altitude-height cross-sections at 120°W on 23 September 2012 cutting the filament over northern America is shown in Fig. 6. A double thermal tropopause (30°N – 65°N) is found along the filament (see black dots in Fig. 6) which encloses a tropospheric intrusion centered around 370 K with a vertical extension of up to 50 K. Air masses associated with a tropospheric intrusion, namely tropospheric air masses intruding polward into the lower stratosphere, are characterized by lower values of ozone, PV, and $N^2$ and higher values of water vapor and CO compared to the stratospheric background (Pan et al., 2009; Vogel et al., 2011b; Ploeger et al., 2013). Indeed, the air mass in the tropospheric intrusion shows low $N^2$ value (Fig. 6, middle right) and enhanced simulated CO (Fig. 6, bottom). For illustration, the 4 PVU and 10 PVU isolines are shown as thick white line in Fig. 6 indicating the climatological isentropic transport barrier at 350 K and 400 K, respectively, for the Northern Hemisphere for September (Kunz et al., 2015).

Enhanced percentages of the emission tracers for India/China (up to ≈ 30%) and Southeast Asia / tropical Pacific Ocean (up to ≈ 33%) are found between the double tropopauses (see Fig. 6, top) compared to the stratospheric background. This demonstrates the horizontal isentropic transport of young tropospheric air masses from the filament into the lower stratosphere. The impact of the emission tracer of the residual surface is less than 10% indicating a minor influence from young air masses originating in other regions of the Earth's surface on the tropospheric intrusion (see Fig. 6 middle left). Thus the transport of tropospheric air masses from the troposphere into the lower stratosphere occurs in the region between the double tropopauses. Previous studies also found intrusions of tropospheric air into the lower stratosphere associated with extratropical double tropopauses (e. g. Pan et al., 2009; Homeyer et al., 2011; Vogel et al., 2011b; Schwartz et al., 2015; Wu and Lü, 2015). Further, it was shown that double tropopauses are frequently associated with Rossby wave breaking events along the subtropical jet (e. g. Vaughan and Timmis, 1998; Castanheira and Gimeno, 2011; Ungermann et al., 2013; Homeyer and Bowman, 2012; Homeyer et al., 2014).



Consequently thin filaments with enhanced contributions of emission tracers for India/China (also for Southeast Asia / tropical Pacific Ocean but not shown here) are found at the polar side of the 7.2 PVU isoline representing the climatological isentropic transport barrier at 380 K in September and October 2012 (see Fig. 7, top, thick black line). These small filaments in the lower stratosphere with enhanced percentages of emission tracers for India/China reach the flight path (red line) of the

TACTS/ESMVal flight on 26 September 2012 over northern Europe. An altitude-height cross-sections at 8°W on 26 September 2012 (marked as thick white line in Fig. 7, top) cross a filament with enhanced contributions of the emission tracers for India/China at 380 K between 60°N and 70°N (Fig. 7, bottom). This filament is clearly found above the first thermal tropopause (black dots) within the lower stratosphere associated with a double tropopause.

Similar filaments within the lower stratosphere were also measured at the flight path of the TACTS/ESMVal flights on 23

and 25 September 2012 (Fig. 8). Subsequently, these filaments are mixed with the stratospheric background and dissipate in the lower stratosphere over time.

### 4.3  Comparison to TACTS/ESMVal measurements

Above, in Sec. 4.1 and 4.2 we considered CLaMS simulations using artificial emission tracers to show that air masses originating in Asia and in the tropical Pacific affect the chemical composition of the lower stratosphere over northern Europe in

September 2012. In this section, we compare results of the three-dimensional CLaMS simulation of CO, $O_3$, $CH_4$, and $H_2O$ with in-situ measurements of three flights conducted on 23, 25, and 26 September 2012 over Europe during the TACTS/ESMVal campaign. The CLaMS results are compared with the in-situ measurements along the flight path by interpolating in time and space. The interpolation method is described in detail in Appendix B.

In Fig. 9 - 11, CLaMS results interpolated along the flight path for each of the three flights are presented. Also shown are

the contributions of CLaMS emission tracers for India/China, Southeast Asia / tropical Pacific Ocean, and the residual surface interpolated along the flight path (see Figs. 9 - 11, top). Further, measured CO, $O_3$, $CH_4$, and $H_2O$ mixing ratios are compared with the CLaMS results. The parts of the flights conducted in the lower stratosphere and characterized by enhanced values of measured CO, $CH_4$, and $H_2O$ and simultaneously reduced $O_3$ compared to the stratospheric background are highlighted in gray.

In general, during all three flights, air masses in the lower stratosphere over Europe are affected by boundary emissions from India/China and Southeast Asia / tropical Pacific Ocean. Contributions up to ≈ 23% and ≈ 25% for emission tracers for India/China and Southeast Asia / tropical Pacific Ocean are found. In contrast, the contribution of the emission tracers for the residual surface are below 7% in the lower stratosphere. Higher contributions are only found in flight parts conducted in the troposphere, e. g. during take off, landing, or flight pattern with a steep decent down into the troposphere followed by a steep

ascent back into the lower stratosphere, referred to as "dive" (see Fig. 9 (top) shortly after 13.00 UTC and Fig. 11 (top) around 10:00 UTC).

In regions with measured enhanced values of measured CO, $CH_4$, and $H_2O$ and reduced $O_3$, generally also the contributions of the emission tracers for India/China and Southeast Asia / tropical Pacific Ocean simulated with CLaMS are enhanced (marked in gray). Further, a good overall agreement between measurements of CO, $H_2O$, $O_3$ and CLaMS simulations is





found. The simulated values of $CH_4$ are in general lower than the measurements (up to $\approx 50\,ppbv$), however the simulated $CH_4$ signatures correspond to the measurements. Differences between model and measurements are most likely caused by uncertainties in the lower boundary conditions of CLaMS (see Sec. 3) which do not include individual emission events of CO or $CH_4$ (see Pommrich et al., 2014).

In the following, we discuss individual signatures of tropospheric air notable in the measurements. These signatures are marked in gray and for clarification, are numbered for each flight (see Fig. 9 - 11, top).

On 26 September 2012 (see Fig. 9, top), a very pronounced signature of tropospheric air in the lower stratosphere is found between 09:05 UTC to 10:17 UTC (No. 2). Here, the contributions of the emission tracer for India/China and Southeast Asia /tropical pacific Ocean are up to 20 % and 23 %, respectively. This is consistent with backward trajectories calculations reported
by Vogel et al. (2014) showing that these air masses are affected by the Asian monsoon anticyclone. Some of the trajectories (2%) originate in the West Pacific and experienced very rapid uplift in typhoon Bolaven on 24/25 August 2012. These air masses reach within 5 weeks the lower stratosphere over Europe.

Further, on 26 September 2012 (see Fig. 9, top), a second pronounced signature of tropospheric air (measured CO, $O_3$, $CH_4$, and $H_2O$ values are of similar magnitude as for No. 2) is found in the lower stratosphere between 08:05 UTC to 10:23 UTC
(No. 1). Also here, the contribution of the emission tracers for India/China and Southeast Asia / Tropical Pacific Ocean are enhanced by up to 10 % and 15 %, respectively. 60-day backward trajectories calculated by Vogel et al. (2014) show that some of these trajectories originate also in the West Pacific region in typhoon Haikui on 2/3 August 2012, however these trajectories need a longer time ($\approx 8$ weeks) from their origin in the West Pacific to reach northern Europe.

During the second part of the flight on 26 September 2012 (see Fig. 9, top), further signatures of tropospheric air are found
(No. 3-6), with enhanced percentages of the emission tracers for India/China and Southeast Asia / tropical Pacific Ocean up to 18 %.

During the flights on 25 and 23 September 2012 (see Figs. 10 and 11) also several lower tropospheric signatures are found in the lower stratosphere. Also here, measured tropospheric signals agree in general with results of the CLaMS model. The results of all these flights confirm that air masses with enhanced amounts of tropospheric trace gases measured in the lower
stratosphere over northern Europe originating in boundary source regions in India/China and Southeast Asia / tropical Pacific Ocean. Our simulations in agreement with measurements show that the amount of water vapor and pollution in the lower stratosphere is enhanced in the Northern Hemisphere in September 2012 associated with the dynamic of the Asian monsoon anticyclone.

## 4.4 Impact from Asian boundary source regions on extratropical lower stratosphere

### 4.4.1 Transport pathways into the lower stratosphere

In the previous sections, the long-range transport pathway from the Asian monsoon anticyclone into the extratropical lower Northern Hemisphere was discussed using a case study. Here, the impact of this horizontal transport on the composition of the





extratropical lower stratosphere is calculated. Fig. 12 shows the horizontal distribution at 380 K of mean values of the emission tracer for India/China for July, August, and September 2012. The temporal evolution of the long-range transport pathway from the region of the Asian monsoon anticyclone into the extratropical lower stratosphere is evident. The frequent separation of air masses at the northeast flank of the Asian monsoon anticyclone and subsequent eastward transport along the subtropical jet is the most important mechanism for long-range transport into the lower northern stratosphere. This long-range transport pathway is most pronounced in September 2012 when the strong eddy shedding event took place as discussed in Sec. 4.1. In Summer 2012, two main horizontal transport pathways of air masses from the Asian monsoon anticyclone with high amounts of the emission tracer for India/China into the TTL evolve (see Fig. 12d):

1. northeastwards along the subtropical jet (eastward eddy shedding and separations of filaments) and the subsequent transport most likely by Rossby wave breaking events into the lower northern hemispheric stratosphere and

2. southwestwards into the tropics (westward eddy shedding) and subsequent mixing within the TTL.

Transport from the Asian monsoon anticyclone both to the east and to the west yields an increase of fresh emissions from India/China within the TTL at 380 K in summer and autumn 2012. Finally, Fig. 12 shows that transport from air masses originating in the Asian monsoon anticyclone into the lower stratosphere take place most likely by Rossby wave breaking over the Pacific and Atlantic Ocean (see Fig. 12c) which is in agreement with findings by Kunz et al. (2015). In our simulation, the horizontal long-range transport pathway along the subtropical jet is furthermore found at levels of potential temperature between 340 K and 420 K (not shown here).

Figs. 10 and 11 show that also air masses originating in Southeast Asia and in the tropical Pacific contribute to the composition of the lower stratosphere over Northern Europe. The horizontal distribution of the emission tracers of Southeast Asia and the tropical Pacific at 380 K from July until September 2012 are shown and discussed in Appendix A to this paper.

To demonstrate that these both transport pathways are also evident in observations, we analyze global HCFC-22 measurements of the MIPAS instrument onboard the ENVISAT satellite (Chirkov et al., 2016). The production of the ozone-depleting and greenhouse gas HCFC-22 ($CHClF_2$) is regulated by the Montreal Protocol and its amendments and adjustments. In accordance with these regulations, in the last decades in some regions e.g. in Eastern Asia and in the Near East, HCFC-22 has been used as interim replacement gas for more potent ozone-depleting substances such as CFCs, although it has been phased out in the developed countries (Fortems-Cheiney et al., 2013). Because HCFC-22 is emitted in locally restricted regions, in particular in Eastern Asia and therefore in the Asian monsoon region, this trace gas is very well suited to study transport processes in the Asian monsoon anticyclone. The region where HCFC-22 emissions occur in Eastern Asia overlap in parts with the emission tracer for India/China in our CLaMS model simulation for 2012.

For summer 2012 no MIPAS measurements are available. Therefore, we compare our results with MIPAS measurements for 2008, because MIPAS HCFC-22 measurements have a very good coverage in summer 2008 over Asia. For this comparison, we perform a CLaMS model simulation for summer 2008 with the same setup for emission tracers as for the model simulation of 2012 described in Sect. 3. Fig. 13a shows mean values of HCFC-22 (Chirkov et al., 2016) measured by the MIPAS for July,





August and September (JAS) 2008 at 380 K. To improve the measurement density of HCFC-22, synoptically interpolating of multiple days of measurements is used through CLaMS trajectory calculations. For each day in JAS 2008, trajectories were computed from the time of measurements in a time window of 5 days (i.e. $-2$ and $+2$ days) to 12:00 UTC of the selected day. Subsequently, the mean HCFC-22 values are calculated on a $3° \times 3°$ latitude-longitude grid between 370 and 390 K.

Similar patterns of HCFC-22 and the emission tracer for India/China are found (see Figs. 13) in the region of the Asian monsoon anticyclone in 2008. This indicates that the two horizontal transport pathways from the Asian monsoon anticyclone to the northeast and to the southwest found in our model results for the India/China tracer in 2012 are also apparent in the MIPAS HCFC-22 measurements and CLaMS simulations for 2008.

  The detailed position of the highest tracer values are somewhat different between 2008 and 2012, probably caused by the
interannual variability of the monsoon. Further, enhanced HCFC-22 values are also found in the Southern Hemisphere at the southern edge of the TTL most likely caused by upward transport in the tropics (see patterns of emission tracer for Southeast Asia and the tropical Pacific in Appendix A). Nevertheless, MIPAS HCFC-22 measurements demonstrate that the large-scale transport pathways from the Asian monsoon anticyclone at its northeastern flank and at its western flank found by artificial tracers of air mass origin in the CLaMS model are also evident in global satellite measurements.

**4.4.2 Flooding of the extratropical lower stratosphere**

The accumulation of young air masses from Asia since 1 May 2012 in the extratropical lower stratosphere is calculated using the isentropic transport barrier at different levels of potential temperature derived by Kunz et al. (2015) as shown in Fig. 14 and Tab. 2. Mean values for different emission tracers are calculated for PV values larger than those at the transport barrier (see Tab. 2) and for air masses poleward of $30°$ N. End of October 2012, the contributions of all boundary emission tracers on
the the composition of the extratropical northern lower stratosphere are at 360 K $\approx$44%, at 380 K $\approx$35%, and at 400 K $\approx 23\%$, with highest contributions from India/China uplifted within the Asian monsoon anticyclone, from Southeast Asia, and from the tropical Pacific Ocean. The contribution of all other regions of the Earth's surface (residue surface) are of minor importance (Tab. 2).

  An equivalent analysis for the Southern Hemisphere (see Fig. 14c and Tab. 2) shows that the contribution of young air
masses on the composition of southern extratropical lower stratosphere is much lower. End October 2012, the contributions of all boundary emission tracers on the composition of the extratropical southern hemisphere lower stratosphere are at 340 K $\approx$27%, at 360 K $\approx$21%, and at 380 K $\approx 2\%$. Here, highest contributions are from tropical Pacific, Southeast Asia, and a minor fraction from India/China representing air masses from the Asian monsoon anticyclone. Also here, the contribution of all other regions of the Earth's surface (residue surface) are of minor importance (see Tab. 2).

Our findings demonstrate the importance of the large-scale Asian monsoon system for the transport of young air masses from Asia and the tropical Pacific into the lower stratosphere of the Northern Hemisphere. This is in particular important for chemical species with lifetimes longer than a few months in the stratosphere such as CO. In agreement with our simulations, during TACTS/ESMVal an increase of the concentrations of long-lived tropospheric source gases (lifetimes $> 3$ months) such as CO, $H_2O$, and $N_2O$ was found by in-situ aircraft measurements in the lower stratosphere over Europe from August until



September associated with transport from the Asian monsoon anticyclone (Müller et al., 2015). These transport processes result in an accumulation of young air masses originating in boundary regions from Asia and the tropical Pacific Ocean within the lower stratosphere in the Northern Hemisphere during autumn 2012.

Because water vapor is an important greenhouse gas and even small perturbations of water vapor mixing ratios in the ExUTLS have a significant impact on surface climate, we are interested to estimate the impact of the Asian monsoon on moistening the lower stratosphere. From our simulations, we roughly estimate the fraction of $H_2O$ originating in India/China, Southeast Asia, and tropical Pacific Ocean contributing to the water budget in the lower northern hemisphere stratosphere. Fig. 15 shows the mean water vapor content in the northern lower stratosphere for PV values larger than 7.2 PVU and northward of 30° N at 380 K calculated with CLaMS (black line). In CLaMS at this altitude, contributions of cirrus clouds to the total water content are of minor importance. An increase of $H_2O$ in the northern lower stratosphere is found in our simulation during summer and autumn as reported in previous studies (e.g. Ploeger et al., 2013; Zahn et al., 2014; Müller et al., 2015). The fraction of $H_2O$ from different boundary tracers is indicated by different colors. End of October 2012, approximately 1.5 ppmv $H_2O$ are from source regions in Asia and the tropical Pacific compared to a mean water vapor content of $\approx 5$ ppmv. Total $H_2O$ without contributions of all boundary tracers (see Fig. 15, green line) shows a decrease during summer and autumn.

Thus, the increase of water vapor in the lower northern stratosphere during summer and autumn can be explained by transport of young tropospheric air masses from Asia and the Pacific ocean affected by the Asian monsoon anticyclone.

We refer this process as 'flooding' of the lower stratosphere with young tropospheric air masses. In previous papers (e.g. Hegglin and Shepherd, 2007; Müller et al., 2015) also the expression 'flushing' as been used. Using the word 'flooding' we emphasize that the northern lower stratosphere is flooded with wet tropospheric young air masses from Asia and the tropical Pacific.

## 5 Discussion

In this paper, the transport mechanisms are analyzed of air masses from inside the Asian monsoon anticyclone to the northern hemisphere lower stratosphere. The combination of separation of anticyclones or filaments at the northeastern flank of the Asian monsoon anticyclone and subsequent horizontal transport along the subtropical jet associated with Rossby wave breaking is put forward here as a long-range transport mechanism from the Asian monsoon anticyclone to the northern hemisphere lower stratosphere. Our model simulations show that air masses originating in Southeast Asia and in the tropical Pacific Ocean uplifted outside of the core of the Asian monsoon anticyclone also contribute to air masses with tropospheric character found in the northern lower stratosphere. This is consistent with 40-day backward trajectory calculations for the TACTS flight on 26 September 2012 (region No. 2 in Fig. 9) showing that about 39% of the trajectories originating in Southeast Asia, Western Pacific and the Asian monsoon region (Vogel et al., 2014). The residual 61% of the trajectories are from the background lower stratosphere. However in the 3-dimensional CLaMS simulations used here, irreversible mixing processes are additionally considered and a time period of 5 months (from 1 May 2012 until September 2012) is simulated which is longer than the 40-day



backward trajectory calculations.

Our model simulation is driven by ERA-Interim reanalysis data. Convection in CLaMS is represented by vertical velocities in ERA-Interim reanalysis data. Thus, small-scale rapid uplift in convective cores is not included in CLaMS simulations. However, previous studies using 3-dimensional CLaMS simulations or trajectory calculations (e.g. Ploeger et al., 2010, 2015; Pommrich et al., 2014; Vogel et al., 2014, 2015; Müller et al., 2015; Ungermann et al., 2016; Konopka et al., 2016) in comparison with satellite or in-situ measurements show that ERA-Interim data are well suited to study transport processes in the vicinity of the Asian monsoon anticyclone and in the tropical tropopause layer.

For the time period of the eddy shedding event discussed in Sect. 4.1, our model study shows that the thermal tropopause is a strong transport barrier above the separated anticyclone for young air masses (younger than 5 months) originating in Asia and the tropical Pacific Ocean. A mixing layer around the thermal tropopause up to $\approx 450\,\mathrm{K}$ is found for fresh emissions (see Fig. 5). The CO simulations also show that the vertical extension of the mixing layer above the thermal tropopause reaches $\approx 450\,\mathrm{K}$ (see Fig. 5) assuming typical CO mixing ratios found in the extra-tropical UTLS ($\approx 30\,\mathrm{ppbv} < \mathrm{CO} < 60\,\mathrm{ppbv}$) (e.g. Hoor et al., 2004; Pan et al., 2004; Vogel et al., 2011b).

Ungermann et al. (2016) analyzed an eastward eddy shedding event in August 1997 observed by the CRISTA space-shuttle mission using PAN (peroxyacetyl nitrate) measurements as a tracer of tropospheric air. They demonstrate that high-resolution CRISTA PAN measurements are very well suited to separate air masses of both the main Asian monsoon anticyclone and of the eastwards separated anticyclone from the stratospheric background. Above the separated eddy, CRISTA measurements also indicate a mixing layer 1-2 km above the thermal tropopause (up to $\approx 420\,\mathrm{K}$) with enhanced PAN mixing ratio a few days after the separation of the eddy from the main Asian monsoon anticyclone. Thus, both our CLaMS simulations for 2012 presented in this paper and PAN CRISTA measurements from August 1997 byUngermann et al. (2016) indicate the existence of a mixing layer above the thermal tropopause of a separated eddy a few days after the eddy shedding event.

# 6 Conclusions

Our findings show that the most important pathways for long-range transport of air masses from the Asian monsoon anticyclone to the extratropical lower stratosphere are eastward migrating anticyclones breaking off a few times each summer from the main anticyclone (Dethof et al., 1999; Popovic and Plumb, 2001; Garny and Randel, 2013; Vogel et al., 2014; Ploeger et al., 2015) and filaments separated on the northeastern flank of the anticyclone. In our case study for the eddy shedding event on 20 September 2012, enhanced contributions of young air masses (younger than 5 months) are found within the separated anticyclone and filaments within the upper troposphere. Within the separated anticyclone, air masses with enhanced tropospheric tracers such as CO and enhanced values of artificial tracers of air mass origin are confined at the top by the thermal tropopause; only a small mixing layer around the tropopause is found in our model simulations. At the polar side, the subtropi-





cal jet acts as horizontal transport barrier. Therefore, the spatial structure of enhanced emission tracers from India/China looks like bubble within the upper troposphere. Those air masses are transported eastwards along the subtropical jet and can enter the extratropical lower stratosphere most likely driven by Rossby wave breaking events associated with double tropopauses (e. g. Vaughan and Timmis, 1998; Castanheira and Gimeno, 2011; Ungermann et al., 2013; Homeyer and Bowman, 2012; Homeyer

et al., 2014). Our simulations show that after entering the lower stratosphere, these air masses are mixed irreversibly with the surrounding stratospheric air. During the TACTS/ESMVal campaign in August and September 2012 conducted with the German Research Aircraft HALO, filaments with enhanced amounts of tropospheric trace gases such as $CO$, $CH_4$, and $H_2O$ and reduced amounts of the stratospheric trace gases $O_3$ compared to the stratospheric background were measured during three flights on 23, 25, and 26 September 2012 over northern Europe within the extratropical lower stratosphere. Our simulations

confirm that these signatures are remnants from an eastwards migrating anticyclone and filaments released at the northeast flank of the Asian monsoon anticyclone that are transported within $\approx 8-14$ days to northern Europe.

In addition to this main transport pathway from the Asian monsoon anticyclone to the east along the subtropical jet and subsequent transport into the northern lower stratosphere, a second horizontal transport pathway out off the anticyclone to the west into the tropics (TTL) is found in agreement with MIPAS HCFC-22 measurements. This second transport pathway is

mainly caused by westward eddy shedding, however this transport pathway yield predominantly to increase of fresh emission from India/China within the TTL at 380 K in summer and autumn 2012 (see Fig. 16).

Moreover, our simulation shows that young air masses originating in Southeast Asia and the tropical Pacific Ocean also contribute substantially to the composition of the lower stratosphere over Northern Europe in summer and autumn 2012. These air masses are spread out during summer and autumn globally on the tropical side of the subtropical jet stream of the Northern

Hemisphere (see Appendix A). A large scale movement of those air masses around the Asian monsoon anticyclone is found in our model simulations indicating that the Asian monsoon anticyclone acts as a large stirrer within the TTL causing mixing of young air masses from Southeast Asia and the tropical Pacific Ocean into the TTL. Subsequent transport into the northern lower stratosphere most likely by Rossby wave breaking events yield to increase of young air masses within the lower northern stratosphere.

Our findings demonstrate that emissions from India/China and Southeast Asia, and tropical Pacific Ocean affected by the circulation of the Asian monsoon anticyclone have a significant impact on the chemical compositions of the lower stratosphere of the Northern Hemisphere by flooding the ExUTLS with young wet air masses in particular at the end of the monsoon season in September/October 2012 in contrast to the southern hemisphere. Our model simulations show that transport from boundary sources in Asia and the tropical Pacific during summer and autumn yield moistening of the the lower northern stratosphere (of

roughly $\approx 1.5$ ppmv end of October 2012) which could have a potential impact on surface climate.

*Acknowledgements.* We sincerely thank Andreas Engel and Harald Bönisch (University of Frankfurt) for coordinating the TACTS campaign, Hans Schlager (DLR Oberpfaffenhofen) for coordinating the ESMVal campaign, the HALO crew, and the HALO pilots. The authors gratefully acknowledge Laura Pan (NCAR, Boulder) and Eric Jensen (NASA Ames Research Center, Moffett Field) for helpful discussions





on transport pathways in the Asian monsoon circulation and their impact on moistening the lower stratosphere. In particular, Laura Pan for comments on the concept of the Asian Monsoon anticyclone as a "tropospheric bubble". We thank the European Centre for Medium-Range Weather Forecasts (ECMWF) for providing meteorological analyses and the ERA-Interim reanalysis data. The authors gratefully acknowledge the computing time granted on the supercomputer JUROPA at the Jülich Supercomputing Centre (JSC) under the VSR project ID

5   JICG11. Our activities were partly funded by the German Science Foundation (Deutsche Forschungsgemeinschaft) under the project LASSO (HALO-SPP 1294/GR 3786) and by the European Community's Seventh Framework Programme (FP7/2007-2013) under the project Strato-Clim (grant agreement no. 603557).



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





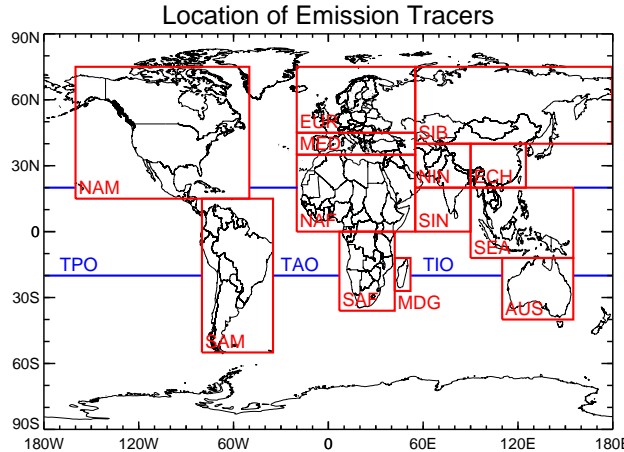

**Figure 1.** Global geographic location of artificial boundary layer sources regions in the CLaMS model, also referred to as 'emission tracers' (Vogel et al., 2015). The latitude and longitude range for each emission tracer is listed in Table 1.

**Table 1.** Latitude and longitude range of artificial boundary layer sources in the CLaMS model, also referred to as "emission tracers". The geographic position of each emission tracer is shown in Fig. 1.

| Emission tracer | Latitude | Longitude |
|---|---|---|
| Northern India (NIN) | 20–40° N | 55–90° E |
| Southern India (SIN) | 0–20° N | 55–90° E |
| Eastern China (ECH) | 20–40° N | 90–125° E |
| Southeast Asia (SEA) | 12° S–20° N | 90–155° E |
| Siberia (SIB) | 40–75° N | 55–180° E |
| Europe (EUR) | 45–75° N | 20° W–55° E |
| Mediterranean (MED) | 35–45° N | 20° W–55° E |
| Northern Africa (NAF) | 0–35° N | 20° W–55° E |
| Southern Africa (SAF) | 36° S–0° N | 7–42° E |
| Madagascar (MDG) | 27–12° S | 42–52° E |
| Australia (AUS) | 40–12° S | 110–155° E |
| North America (NAM) | 15–75° N | 160–50° W |
| South America (SAM) | 55° S–15° N | 80–35° W |
| Tropical Pacific Ocean (TPO) | 20° S–20° N | see Fig. 1 |
| Tropical Atlantic Ocean (TAO) | 20° S–20° N | see Fig. 1 |
| Tropical Indian Ocean (TIO) | 20° S–20° N | see Fig. 1 |
| Background | remaining surface | |

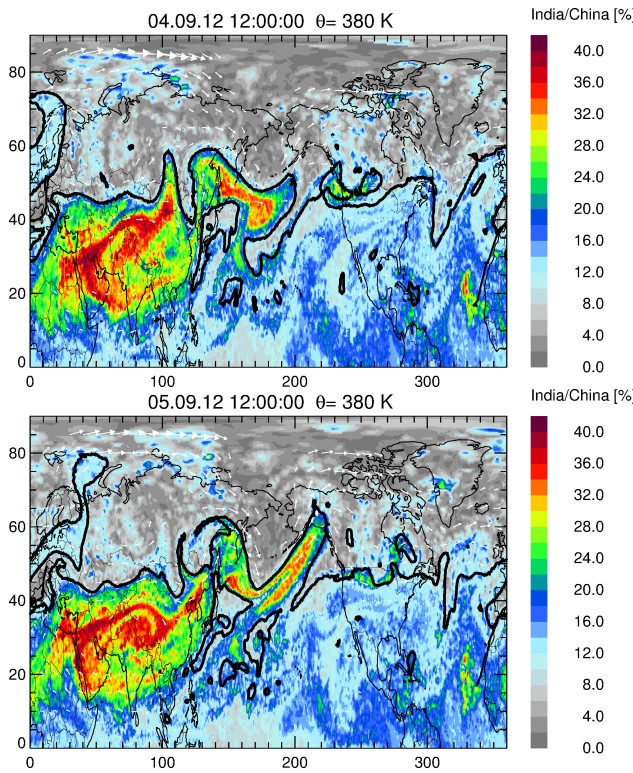

**Figure 2.** Horizontal distribution of the fraction of air originating in India/China (here the sum of emission from North India, South India, and East China) at 380 K potential temperature on 4 (top) and 5 (bottom) September 2012. The horizontal winds are indicated by white arrows. The 7.2 PVU surface is shown as thick black line indicating the climatological isentropic transport barrier at 380 K in September (Kunz et al., 2015).

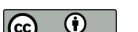

**Figure 3.** Horizontal distribution of PV (left) and the fraction of air originating in India/China (right) (here the sum of emission from North India, South India, and East China) at 380 K potential temperature on 20, 23, and 26 September 2012. The horizontal winds are indicated by white arrows. The 7.2 PVU surface is shown as thick black line indicating the climatological isentropic transport barrier at 380 K in September (Kunz et al., 2015). The the thick white lines at 180°E and at 120°W mark the position of vertical curtains shown in Figs. 5 and 6.





**Figure 4.** As Fig. 3, but for simulated CO (left) and the fraction of air originating in Southeast Asia / tropical Pacific Ocean.





**Figure 5.** Latitude-height cross-sections from 40°S to 90°N at 180°E longitude on 20 September 2012 including the separated anticyclone (≈ 25 − 45°N) caused by an eddy shedding event for the fraction of air originating in India/China (top left), the fraction of air originating in Southeast Asia / tropical Pacific Ocean (top right), the fraction of air originating in the residual surface (bottom left), and buoyancy frequency squared $N^2$ (bottom right). The climatological isentropic transport barrier of 4 and 10 PVU at 350 K and 400 K, respectively, for September (thick white lines) and thermal tropopause (black dots) are shown. The corresponding levels of potential temperature are marked by thin white lines.





**Figure 6.** Latitude-height cross-sections of the Northern Hemisphere at $120°$W longitude on 23 September 2012 including an northward directed filament of low PV ($\approx 35-80°$N) moving along the subtropical jet for the fraction of air originating in India/China (top left), the fraction of air originating in Southeast Asia / tropical Pacific Ocean (top right), the fraction of air originating in the residual surface (bottom left), and buoyancy frequency squared $N^2$ (bottom right). The climatological isentropic transport barrier of 4 and 10 PVU at 350 K and 400 K, respectively, for September (thick white lines) and thermal tropopause (black dots) are shown. The corresponding levels of potential temperature are marked by thin white lines.





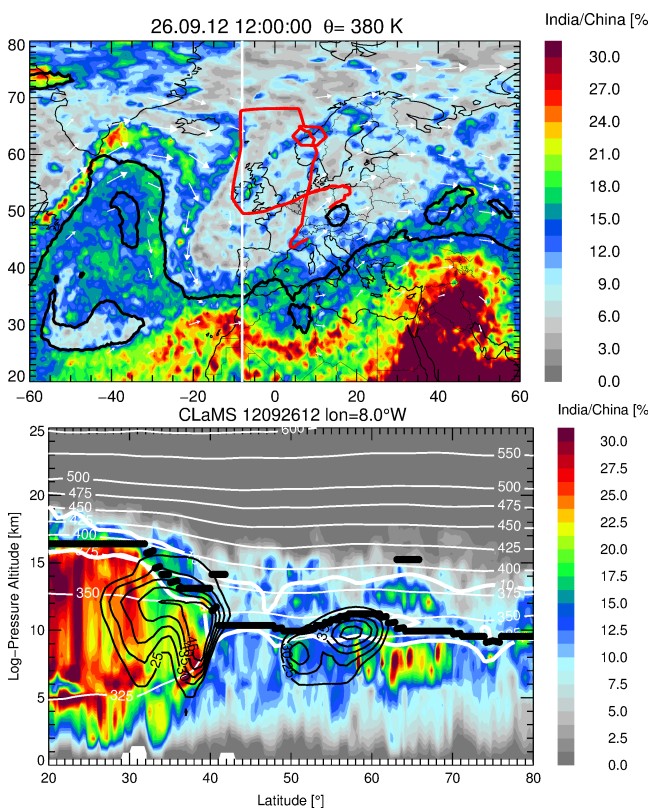

**Figure 7.** Horizontal (top) cross-section of the fraction of air originating in India/China over Europe om 26 September 2012. The flight path transferred to noontime of the TACTS/ESMVal flight is shown as red line. The climatological isentropic transport barrier of 7.2 PVU at 380 K is shown as thick black line. The white thick line marks the position of the vertical (bottom) cross-section at 8°W longitude which is similar to the cross-sections shown in Fig. 5 and 6.

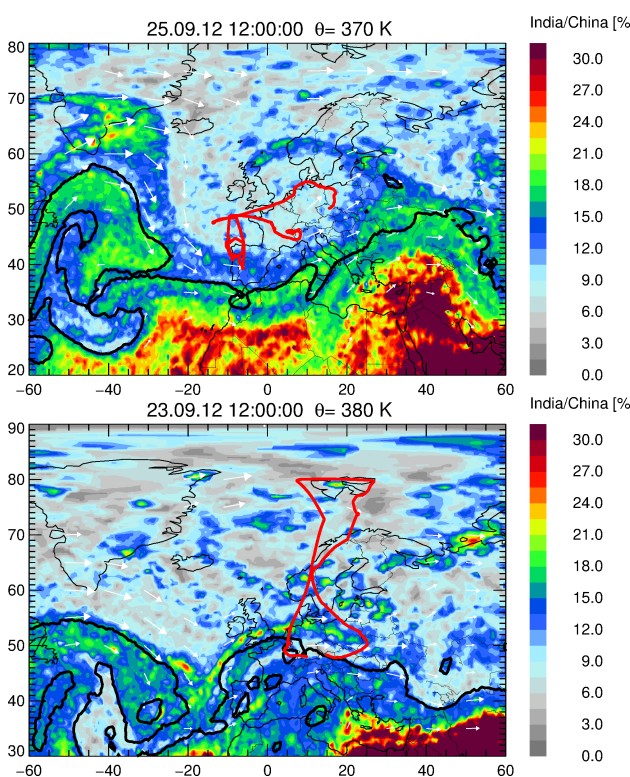

**Figure 8.** Horizontal cross-section of the fraction of air originating in India/China over Europe on 25 September at 370 K (top) and on 23 September 2012 at 380 K (bottom). The flight paths transferred to noontime for the TACTS/EsmVal flights 2012 over northern Europe are marked as red lines. The climatological isentropic transport barrier of 6.0 PVU (at 370 K) and 7.2 PVU (at 380 K) are shown as thick black lines.







**Figure 9.** Top Panel: Time evolution (given in UT time) of potential temperature, the fraction of air originating in India/China, Southeast Asia/tropical Pacific and of the residual surface (sum of all other emission tracers covering the entire Earth's surface) simulated with CLaMS for the flight on 26 September 2012. Middle and Lower panel: CO, $O_3$, $CH_4$, and $H_2O$ simulations and measurements. Segments of the flight in the lower stratosphere with enhanced measured CO, $CH_4$, and $H_2O$ and reduced $O_3$ compared to the stratospheric background are highlighted in gray and numbered for clarification.






**Figure 10.** As Fig. 9, but for the flight on 25 September 2012.





**Figure 11.** As Fig. 9, but for the flight on 23 September 2012.





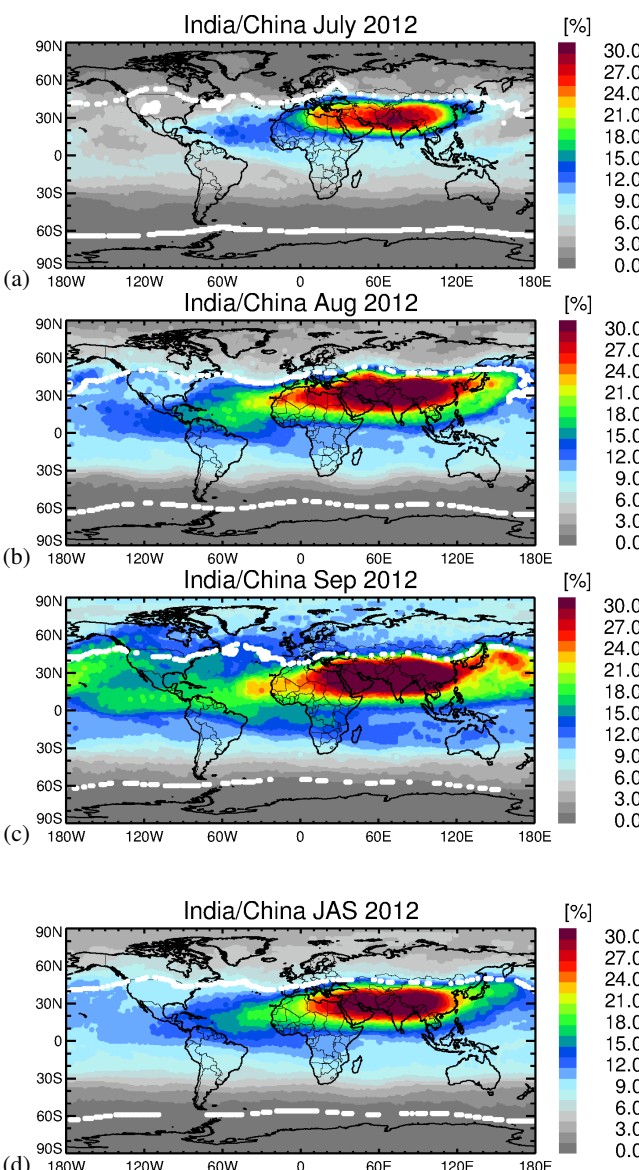

**Figure 12.** Mean values of the contribution of the emission tracer for India/China at 380 K in July (a), August (b) and September (c) 2012. Plate (d) shows mean values for India/China for July, August and September 2012. The climatological isentropic transport barriers of 7.2 PVU (northern hemisphere) and −11.5 PVU (southern hemisphere) at 380 K are shown as thick white dots.



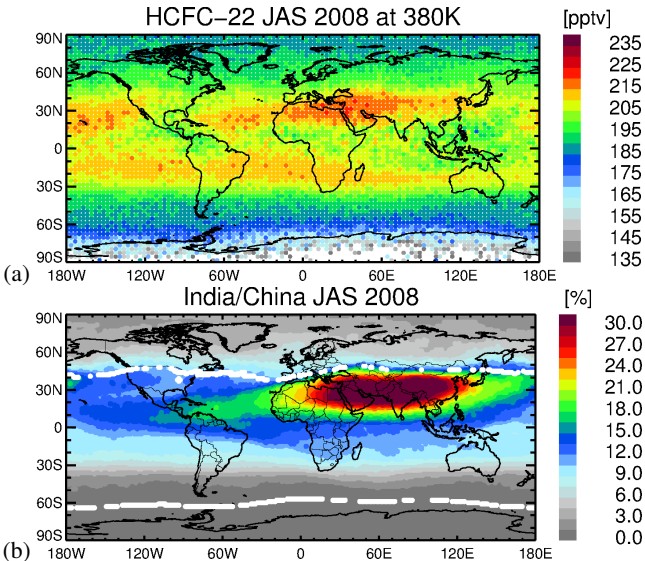

**Figure 13.** Mean values of HCFC-22 (a) measured by MIPAS for July, August, and September 2008 at 380 K synoptically interpolated using CLaMS trajectory calculations (details see Sect. 4.4.1). Plate (b) is similar as Fig. 12d, but for the year 2008.

**Table 2.** The contributions of all boundary emission tracers on the the composition of the extratropical lower stratosphere end of October 2012 at different levels of potential temperature using the climatological isentropic transport barrier derived by Kunz et al. (2015) for the northern and southern hemisphere. Highest contributions are from India/China, Southeast Asia, and the tropical Pacific Ocean. The contribution of all other regions of the Earth's surface are of minor importance (residue surface).

| Θ | transport barrier | contribution of all boundary tracer | contribution of residue surface |
|---|---|---|---|
| Northern Hemisphere | | | |
| 340 K | 3.8 PVU | 48 % | 7 % |
| 360 K | 5.5 PVU | 44 % | 5 % |
| 380 K | 7.2 PVU | 35 % | 4 % |
| 400 K | 10.0 PVU | 23 % | 3 % |
| 420 K | 13.5 PVU | 14 % | 1 % |
| Southern Hemisphere | | | |
| 340 K | −3.0 PVU | 27 % | 3 % |
| 360 K | −5.6 PVU | 21 % | 2 % |
| 380 K | −11.5 PVU | 2 % | < 1 % |

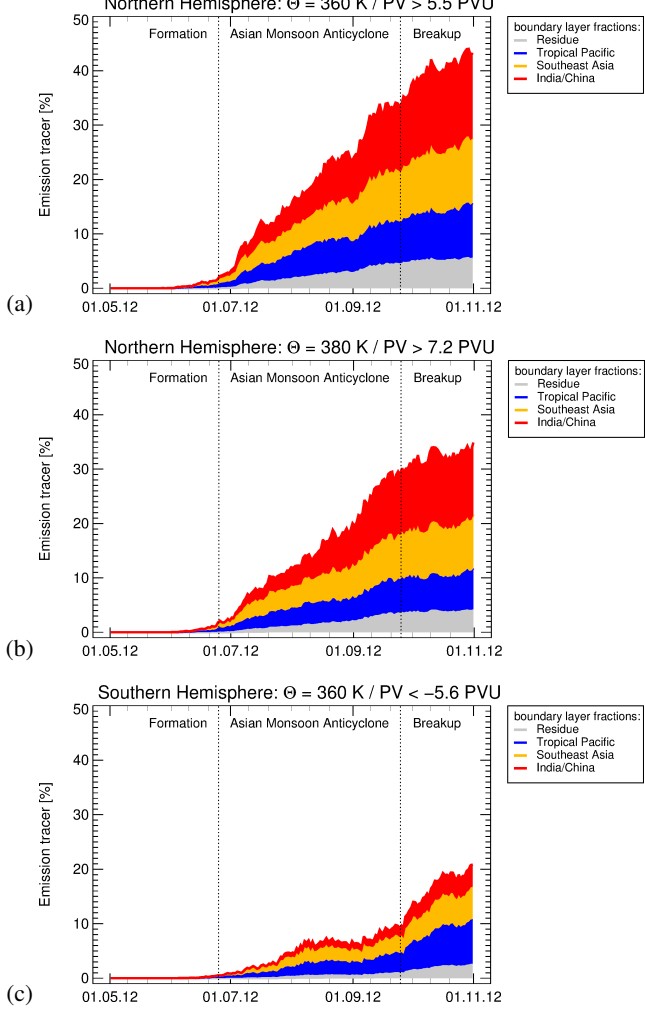

**Figure 14.** Contribution of different emission tracers from India/China, Southeast Asia, tropical Pacific Ocean, and residual surface to the northern lower stratosphere at different levels of potential temperature 360 K (a) and 380 K (b) from May until October 2012. The same for the southern lower stratosphere at 360 K (c).





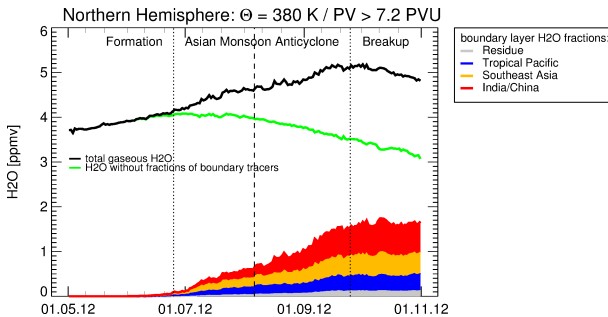

**Figure 15.** The increase of $H_2O$ mixing ratios in the lower northern hemisphere stratosphere at 380 K during summer 2012 is shown (black line). The green line indicated $H_2O$ mixing ratios without fractions from the Earth's boundary layer. A rough estimation of the fraction of $H_2O$ mixing ratios originating in India/China (red), Southeast Asia (yellow) and the tropical Pacific Ocean (blue) in the northern lower stratosphere at 380 K is also given.

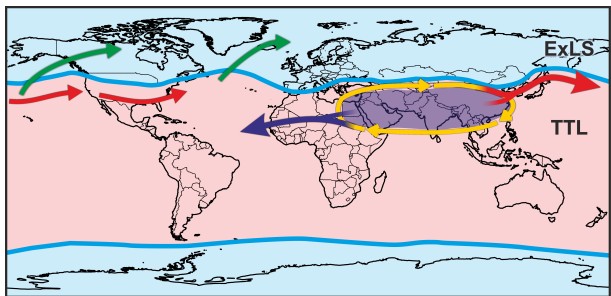

**Figure 16.** Horizontal Transport pathways from the Asian monsoon monsoon anticyclone and the surrounding air masses (yellow) westward into the tropical tropopause layer (blue arrow) and eastward along the subtropical (red arrows). Cross-tropopause transport into the extra-tropical lower stratosphere occurs mainly above the Atlantic and Pacific Ocean most likely driven by Rossby wave breaking events (green arrows).



## Appendix A: Additional emission tracers

### Horizontal distributions of emission tracers for Southeast Asia and the tropical Pacific Ocean

Figs. 10 and 11 show that also air masses originating in Southeast Asia and in the tropical Pacific contribute to the composition of the lower stratosphere over Northern Europe. The horizontal distribution of the emission tracers of Southeast Asia and the tropical Pacific at 380 K from July until September 2012 are shown in Fig. 17. Air masses with enhanced percentages of the emission tracer for tropical Pacific Ocean are found over the Maritime Continent and the western Pacific in July and August 2012 indicating upward transport outside of the Asian monsoon anticyclone (see Fig. 12 a and b). In September 2012, enhanced values are also found westward and eastward of the Maritime Continent indicating horizontal transport within the TTL (see Fig. 17c and d (left)). Low values are found in the regions of the Asian monsoon anticyclone indicating the transport barrier around the anticyclone which inhibit horizontal exchange of air masses. Further transport into the northern lower stratosphere also occurs most likely by Rossby wave breaking events along the subtropical jet similar as for air masses from the Asian monsoon anticyclone (enhanced contributions from India/China; see Fig. 12).

Air masses with enhanced contributions from Southeast Asia are found both within the Asian monsoon anticyclone and over the Maritime Continent and the western Pacific caused by the geographic position of the emission tracer of Southeast Asia as shown in Fig. 17 (right). Thus, enhanced contributions from Southeast Asia are found as a a superposition of transport within the Asian monsoon anticyclone and outside in the western Pacific region is evident in the horizontal distribution of the emission tracer for Southeast Asia.

## Appendix B: Description of interpolation method

### Full 3-D Delaunay triangulation

In this study, the used interpolation method to determine mixing ratios of chemical species and percentages of artificial tracers of air mass origin along the flight path of the TACTS/ESMVal flights (see Figs. 9–11) from CLaMS data is upgraded from the previously used semi-2-D scheme to a full 3-D Delaunay triangulation.

As a first step, CLaMS backward trajectories are calculated from time and space of the measurements to the CLaMS model output at noon one day before the measurement. Subsequently, the CLaMS data calculated on an irregular grid are interpolated on the endpoints of the backward trajectories as described below.

The complexity of creating the Delaunay triangulation increases in the worst-case with the square of the inserted points. To reduce the number of points and to get rid of the spherical shape of the points, it is advantageous to partition the surface of the Earth in six latitudinal bands (0° to 36°, 36° to 72°, and 72° to 90° for each hemisphere). The four latitudinal bands between 72°S and 72°N are in addition split into longitudinal areas (every 36° for the two bands between 0° and 36°; every 45° for the two bands between 36° and 72°). This partitioning was selected to to give roughly evenly sized areas with a sufficient amount



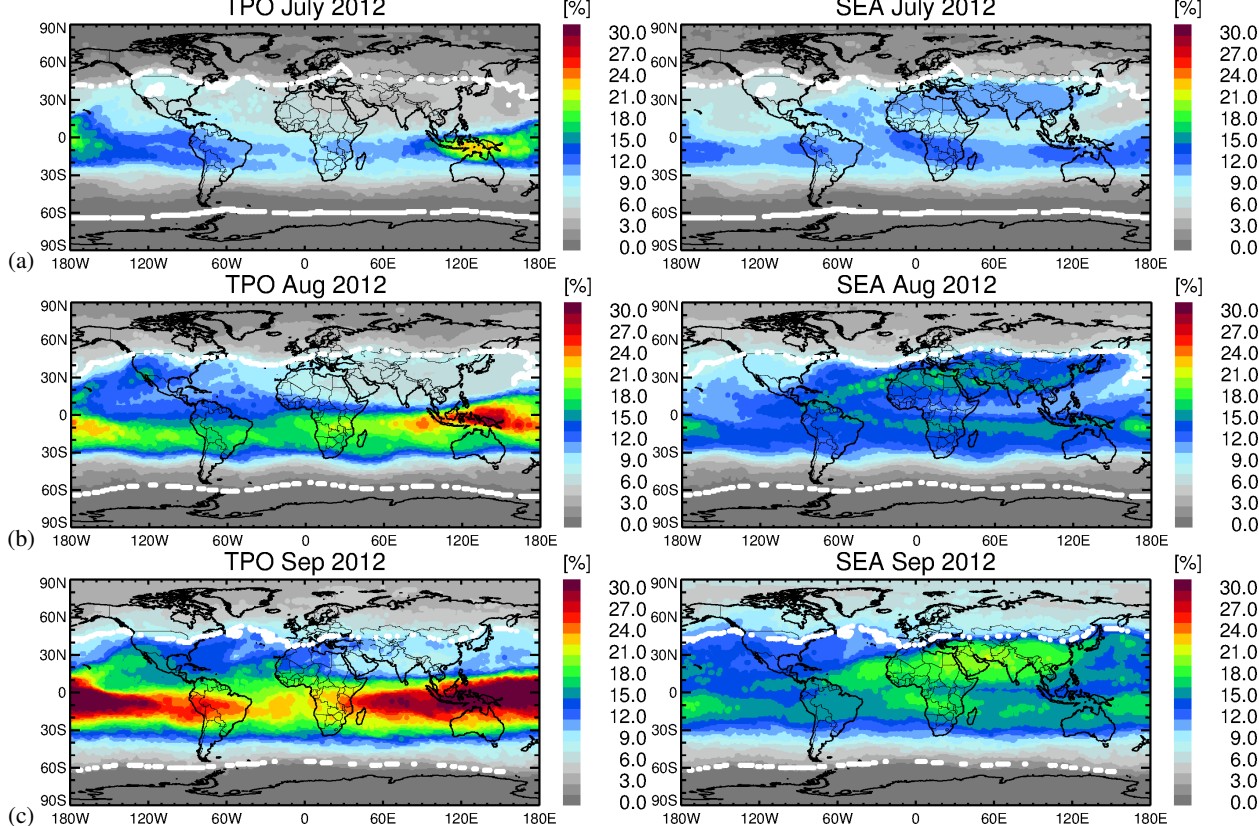

**Figure 17.** Mean values of the contribution of the emission tracer for tropical Pacific Ocean (left) and Southeast Asia (right) at 380 K in July (a), August (b) and September (c) 2012. The climatological isentropic transport barriers of 7.2 PVU (northern hemisphere) and $-11.5$ PVU (southern hemisphere) at 380 K are shown as thick white dots.

of air parcels. Within each of these areas, a Cartesian 3-D Delaunay triangulation including all contained parcels and a certain

5   amount of neighboring parcels (here extending each area by 25% in longitude and latitude) is employed (Delaunay, 1934). The chosen amount of overlap assures that all relevant parcels surrounding the area are included in the triangulation for the typical air parcel density of the examined simulations. The coordinate system employed for each area is locally projected using a stereographic projection on the horizontal center of each area.

     In the CLaMS simulation used here, a horizontal resolution of 100 km is used with a maximum vertical resolution (thickness

10  of the model layer) of about 400 m around the tropopause and thicker layers above and below the tropopause (in general the geometric thickness of a layer $\Delta z$ is described by $\Delta z = \alpha \Delta r$ with the aspect ratio $\alpha$ and the horizontal resolution $\Delta r$; $\alpha$ is controlled by the assumption that the entropy of the system is uniformly distributed over all air parcels (more details see Konopka et al., 2012; Pommrich et al., 2014)). The previously used semi-2-D scheme interpolation is effectively restricted to occur within one model layer.





5    The thickness of each layer in hybrid coordinates is defined corresponding to a fixed amount of a virtual altitude in kilometer $\delta$ within the coordinate system of each area. The vertical position of each air parcel within a layer is assigned to the virtual coordinate by its relative position in the layer: If the boundaries of the layers are defined as $\zeta_i$ with $\zeta_{i+1} > \zeta_i$, $i \in \mathbb{N}_0$, then the virtual vertical coordinate $z'$ of an air parcel with vertical hybrid coordinate $\zeta$ within one area is defined as

10    $$z' = \left( i + \frac{\zeta - \zeta_i}{\zeta_{i+1} - \zeta_i} \right) \cdot \delta \qquad\qquad\qquad (\text{B1})$$

For the CLaMS simulation used here, a factor of $100\,\mathrm{km}$ was chosen for $\delta$, corresponding to the typical horizontal distance of parcels within one layer according to the horizontal resolution.

Not used here, but useful for other applications is a simple linear relationship between the potential temperature of an air parcel and its virtual vertical altitude of, e.g., $z' = \theta \cdot 10\,\mathrm{km/K}$.

Using the coordinate transformation with the virtual vertical coordinate as defined in equation B1 and using the segmentation in different latitude-longitude areas, the triangulation can be calculated in Cartesian coordinates involving a strongly reduced amount of air parcels compared to all air parcels of the model simulation. Using the 3-D triangulation, several fast and useful interpolation options are readily available, such as nearest neighbor or barycentric interpolation. We choose to employ natural
5    neighbor (or Sibson) interpolation (Sibson, 1981), which computes the weighted means of all neighboring points in contrast to methods such as the barycentric interpolation, that use only neighboring points spanning a tetrahedron around the enclosed air parcel.

The Computational Geometry Algorithms Library (CGAL) offers all routines necessary to efficiently implement the described algorithm (Pion and Teillaud, 2013). The algorithm and its variants were implemented as a python module called juregrid3d (Jülich Regridding 3D) using C++ and cython for the performance critical parts.