# Peer review of "Long-range transport pathways of tropospheric source gases originating in Asia into the northern lower stratosphere during the Asian monsoon season 2012"

_Atmospheric Chemistry and Physics, 2016_

## Referee Comment (RC1) · Anonymous Referee #1 · 19 Jul 2016

This paper presents an analysis of the evolution and transport of air with emission sources in the Asian monsoon region to the northern hemisphere extratropical lower stratosphere using primarily output from the Chemical Lagrangian Model of the Stratosphere (CLaMS) driven with winds from the ERA-Interim reanalysis. Aircraft observations are used to establish confidence in the model simulations and demonstrate linkages between transport signatures and air masses observed in the extratopical lower stratosphere downstream. The main conclusions of the study are that the air lofted into the Asian monsoon anticyclone is confined to the tropical upper troposphere during the monsoon season, but is transported poleward into the extratropical lower

stratosphere during the breakup of the Asian monsoon anticyclone in early fall via mostly Rossby wave breaking events over the Pacific and Atlantic ocean basins. The paper is well-detailed, well-structured, and well-reasoned. I do not find any fundamental errors in the analysis or questionable claims in the attendant discussion, but I believe the paper will benefit from a bit more cohesion of arguments throughout and a bit more analysis of the seasonality of transport. I recommend that the paper be considered for publication after mostly minor revisions. My general and specific comments to help guide the revisions are provided below.

**General Comments:**

1. The linkages between transport of the monsoon upper troposphere air to the extratropical lower stratosphere and poleward Rossby wave breaking downstream of the anticyclone are clearly established in the manuscript. However, I feel the authors miss an opportunity to strengthen these points as their analysis expands in the latter part of the paper. Namely, the recent climatological studies of Rossby wave breaking cited in the paper provide support for the observed seasonality and timing of "flooding" of moist monsoon air into the extratropical lower stratosphere and for the dominant locations of poleward wave breaking events pointed out in Figure 16. Such additional detail on these linkages will provide better continuity in the Results section and strengthen the presentation and discussion of the conceptual model given in Figure 16.

2. While the authors do a good job of including observational support for the model results with analysis of a few flights during the TACTS/ESMVal campaign, a more thorough test of the CLaMS model (particularly related to the influence of Asian monsoon air on extratropical lower stratosphere water vapor – i.e., Figure 15) using the same chemical dataset that was used for initialization – Aura MLS – is requested. Is the enhancement in extratropical lower stratosphere water vapor

at 380 K following breakup of the Asian monsoon anticyclone an observed characteristic? I expect this is a straight-forward test of the model and would go a long way in strengthening the perceived impact of this paper. If not an observed characteristic, this is a questionable result.

3. The naming convention for emission sources outside of the Asian monsoon region is inconsistent in the text and Figures of the manuscript. While the use of "residual" is common and seems to be the primary intention of the authors, "residue" appears in other places (e.g., Figures 5, 6, 14, and 15; Table 2; and Page 12, line 23; Page 12, line 29). Please update the text and figures to refer to this as "residual" throughout.

4. While the figures are (for the most part) visually appealing, an effort should be made to have the spacing, scaling, and text sizes consistent throughout. For example, the color bar labels run into the latitude axes in Figures 3 and 4, the cross-sections are misaligned in Figures 5 and 6, text sizes of the two panels in Figure 7 are different, the bottom panel is unnecessarily displaced from the top three in Figure 12, and the text sizes in Figures 14 and 15 are not legible at normal zoom.

**Specific Comments:**

Page 1, line 7: "...jet such as the..." should be "jet such that the"

Page 1, lines 16-17: This statement is confusing here. This should be clarified to say that sources from Asia and the tropical Pacific account for ~1.5 ppmv of the ~5 ppmv mean in the extratropical lower stratosphere.

Page 2, line 3: "...is acting..." should be "...acts..."

[Figure]

Page 3, lines 16-23: This discussion is limited to large-scale transport processes, correct? For example, we know that moist convection (apart from a large organized system like a typhoon) is capable of transporting air across the tropopause but such small-scale processes (though possibly represented to an extent) are not resolved in these models. A bit more detail and context should be given to clarify these points here, which you do reflect on near the end of the paper.

Page 3, line 27: "...are used..." should be "...is used..."

Page 4, line 14: "Measurements of..." should be "Measurements from..."

Page 6, line 30: "...occurs like for all..." should be "...occurs in an equivalent fashion to all..."

Page 7, line 20: "...as for..." should be "...to..."

Page 8, line 12: "...into stratosphere." should be "...into the stratosphere."

Page 8, lines 15-18: The Homeyer et al 2011 paper you cite can be referenced here as well.

Page 10, lines 11-12: What do you mean by this statement? It takes 5 weeks for the parcels to be transported from their surface emission locations to the lower stratosphere over Europe? This statement needs to be clarified a bit.

Page 10, lines 19-21: Based on the time series, No. 3 is an aircraft-only signature (i.e.,

no apparent plume in the CLaMS simulation – at least not to me!).

Page 10, lines 26-18: But is this really dynamics of the AMA or of a downstream RWB event? You have already demonstrated that the latter is the reason this particular air mass crossed the tropopause, correct?

Page 10, line 35: "Northern Hemisphere" should be "stratosphere"

Page 11, line 22: Remove "these"

Page 12, line 25: "...masses on the..." should be "...masses to the...", and "End October..." should be "End of October..."

Page 12, line 27: "Here, highest contributions are from tropical..." should be "Here, the highest contributions are from the tropical..."

Page 12, line 31: "This is in particular..." should be "This is particularly..."

---

## Referee Comment (RC2) · Anonymous Referee #2 · 28 Jul 2016

Summary: In this study the authors examine the transport characteristics associated with the Asian summer monsoon during September-October 2012, using both measurements of trace gases (e.g. ozone, water vapor, methane) and idealized tracer simulations that provide information about the relative contributions of different boundary layer regions to upper tropospheric/lower stratospheric air masses. The study provides strong evidence that the eastward shedding of eddies from the monsoon anticyclone provides an important mechanism for transporting boundary layer air from India and South Asia to the lower stratosphere over northern midlatitudes. While this study provides a comprehensive analysis and important contribution to the field that will make

it suitable for publication, I have a few major comments that need addressing before I recommend its publication. In particular I have one major concern about the authors' interpretation of the air-mass origin tracers that needs addressing, as it may potentially affect the interpretation of the main results. I also have smaller comments that are, by comparison, less important.

Major Comment:

1) I am concerned about the interpretation of the air-mass tracers as a fraction. It is definitely constructive to look at the relative contributions of different source regions and I commend the authors' use of the diagnostic. However, more care should be taken in the interpretation of the tracer concentration as giving the fraction of air that was last at the earth's surface in a given source region. In particular, the simulation only covers 1 May 2012 - 31 October 2012. If the tracers are to be interpreted as fractions (as the authors intend them to be) then the sum of the air-mass tracers corresponding to all of the source regions must equal 1 (since the union of the source regions is the entire planetary boundary layer (PBL)). This is not the case, however, as shown in Figure 9. The sum of the red, blue and orange lines should, in principle, equal 1 (but does not). What this tells me is that the tracers have not been integrated to equilibrium so that there is a large amount of air that is not accounted for by the source fractions. This is a known issue when dealing with air-mass origin tracers (Orbe et al. (2015)) and I am concerned about what this means for the main conclusions in the study. Please either start the simulation much earlier (to ensure tracer equilibration by September 2012) or remove all references to "fraction" because this interpretation is not correct. Alternatively, it is possible that I am missing something important in the authors' definition of "residual" (by which I interpret the rest of the PBL) – if this is the case, please clarify in the text.

Orbe, Clara, Paul A. Newman, Darryn W. Waugh, Mark Holzer, Luke D. Oman, Feng Li, and Lorenzo M. Polvani. "Airmass Origin in the Arctic. Part I: Seasonality." Journal of Climate 28, no. 12 (2015): 4997-5014. (Figure 3b)

[Figure]

Minor Comments:

1) Line 29, Page 3: I am wary of the use of the term "transport pathways." The air-mass origin tracers only tell you where air was last in contact with the boundary layer. They do not provide a sense for how the air arrived at the receptor location, so please remove all references to pathways. To infer pathways you would need to use idealized tracers similar to those used for inferring the age spectrum or, most appropriately, the path density tracers examined in Holzer (2009).

Holzer, Mark. "The path density of interhemispheric surface-to-surface transport. Part I: Development of the diagnostic and illustration with an analytic model." Journal of the Atmospheric Sciences 66, no. 8 (2009): 2159-2171.

2) Line 8, page 6: Again, reservation about the word "pathway."

3) Lines 33-35, page 6: The sum of all of the air-mass fractions does not equal 1, leaving a large fraction of air unaccounted for. Therefore, I am not confident in the statement that "that air masses originating in India/China and Southeast Asia/Pacific Ocean almost exclusively contribute to the chemical composition of the separated anticyclone." Please either start your simulation earlier (to ensure equilibration of the air-mass tracers) or do not use the word "fraction."

---

## Referee Comment (RC3) · Anonymous Referee #3 · 30 Aug 2016

Review of 'Long-range transport pathways of tropospheric source gases originating in Asia into the northern lower stratosphere during the Asian monsoon season 2012," by Vogel et al., submitted to ACPD, 2016

The authors use the global Lagrangian CLaMS model, with artificial and chemical constituent tracers to quantify the contributions of different boundary Layer (BL) source regions in Asia to the Asian Summer Monsoon (ASM) anticyclone, and from there to extra-tropical lower stratosphere (ExLS), for the 2012 ASM season. Further, they illustrate the transport pathways for BL source air, accumulated in the ASM anticyclone, to reach the ExLS, via eddy shedding in the upper troposphere, subsequent filamentation and penetration into the stratosphere, associated with Rossby wave breaking along the subtropical jet. They also consider the westward shedding of air from the ASM anticyclone into the tropical upper troposphere.

The authors use the simulated artificial tracers, and simulated ozone, CO, and water vapor, to interpret small-scale structures observed along aircraft flight tracks as filamentary intrusions of BL source air associated with the ASM anticyclone into the lower stratosphere over Northern Europe. Further, they use the artificial tracers to *quantify* the contribution of different BL source regions to the ExLS over the 2012 ASM season, and use CLaMS simulated water vapor to estimate the contribution of Asian source regions to water vapor in the ExLS.

The study builds upon earlier studies that have illustrated troposphere-stratosphere exchange (STE) mechanisms associated with the ASM, going back at least as far as Dethof et al. (1999), a paper which the authors cite. At the same time the use of the artificial and constituent tracers with the CLaMS model to *quantify* estimates of the contributions from Asian (and other) BL source regions to the ExLS is I believe a step beyond these earlier studies. Quantification of water vapor contribution to the NH lower stratosphere (p.13) is particularly interesting.

The cross-section of the filamentary structure (Fig. 6) provides an illuminating illustration of intermediate (mixed) constituent and stability conditions between the tropospheric and stratospheric air mass characteristics.

The comparison of aircraft observations of the low stratosphere over Northern Europe with CLaMS tracer maps interpolated to the aircraft flight tracks illustrates effectively that the simulation of tropospheric filaments in the low stratosphere represents real-world conditions, and supports the quantitative estimates of BL source influence in the ExLS presented later. The analysis and interpretation is reminiscent of that conducted by Fairlie et al. (2007) for INTEX-NA aircraft observations; the authors may wish to add correlation scatter plots of the observed O3, CO or CH4, water vapor to further illuminate air mass origin and characteristics of mixed troposphere-stratosphere air masses.

I think the paper could use an editorial review for the English and sentence structure. There is occasional awkwardness in the sentence structure, and some choice of wording that I find confusing, and may be a translation issue (see some examples below).

Nevertheless, I think the paper is suitable for publication in ACP given consideration to these issues and the points listed below, most of which are minor and for the purpose of clarification.

p.6, line 31, What is meant by "Maritime Continent"?

p.7, line 12-13: Comment: The authors will recognize that the transport is only irreversible if the tropospheric intrusion is mixed into the stratospheric surroundings. It is conceivable that an intrusion across the PV=7.2 PVU could return to the troposphere downstream.

p.7, lines 14-15: Comment: I am unable to see the PV=7.2 PVU isopleth enveloping a "region of enhanced tracers"

p.7, line 31, instead of "surface that is" do you mean "surface, i.e.,"? I.e., are the authors stating the definition of "residual" here?

p.7, line 32-33, reference "no signature." Would the authors be more quantitative here? Looks like up to 10-15% is due to "residual" sources in the anticyclone.

p.8, line 12, reference "indicating transport from the troposphere into the stratosphere." This is according to the definition of the authors, based on the work of Kunz et al.

p.9, reference discussion of Fig. 9 emission tracer plots, here and elsewhere. Please confirm for the reader if "residual" includes all BL surfaces other than those identified (China/India, SEAsia/ tropical Pacific). How should the reader interpret the sum of these percentages being much less than 100%, e.g. does the remainder comprise background lower stratospheric air, unconnected to any BL surface in past 5 months?

p. 10, discussion of Figs. 9-11. It would be helpful if the authors labeled the locations of flight segments "1", "2", "3", etc. on the maps in Figs. 7-8, to help the reader identify the features highlighted in the flight data to features on the CLAMS maps.
Additionally, the flight data appears to be higher temporal resolution than the CLAMS profiles (e.g. the profile of FISH H2O). It may be helpful to add a time-averaged data profile at the same resolution as the CLaMS for better comparison. Tracer-tracer correlation plots may also be a useful addition (see above).

p. 11, lines 13-15. Suggestion: I think the authors mean to emphasize the locations (Atlantic and Pacific Oceans) here, rather than the mechanism (Rossby wave breaking). They may want to leave out "Rossby wave breaking" in this sentence to keep the stress on the locations.

p.11, lines 18-20. This sentence seems a bit out of context here. Perhaps reference to SE Asia/ Tropical Pacific contribution (the appendix) would sit better after the introduction to Fig.12 (p.11, line 2, after "September 2012").

p.11, reference discussion of "transport pathways": The title of 4.4.1 is "transport pathways into the lower stratosphere." The authors have illustrated that "eastward eddy shedding" on the NE side of the anticyclone, and subsequent transport and filamentation of material can be a pathway to reach the stratosphere (pathway 1, line 10). But, what about the "westward eddy shedding" (pathway 2, line 11) from the anticyclone to the TTL? I find no discussion of this as a potential pathway to the stratosphere, via e.g. diabatic ascent (Garny and Randel, 2015).

p.12, discussion of Fig. 14 and Table 2. Please clarify how these metrics are computed. Are they achieved by area weighting daily isentropic " fraction of air" maps for areas north of $30^o$N and for PV greater than the "transport barrier" PV?
Are you saying for example that by end October 2012, almost 20% of the air in the NH at 360K north of these delimiters originates in the India/China BL within the previous 5 months?

p.14, lines 12-14, reference "A mixing layer …. (see Fig. 5)" It is unclear to me what feature is being identified in Fig. 5. I see no discussion of such a mixing layer in earlier discussion of Fig. 5. Indeed 2 lines earlier (p.14, line 11) the thermal tropopause is described as a "strong transport barrier above the separated anticyclone." Do you really mean Fig. 5? Are you referring to the thin layer of strong vertical gradients in fractions of air and in simulated CO at the thermal tropopause south of ~$40^o$N, ~370-400K)? What is the evidence for mixing, and what is the mechanism? Or do you mean Fig. 6 instead where mixed troposphere-stratosphere characteristics of BL source fractions, CO, and buoyancy frequency *are* evident between the double thermal tropopauses? The discussion of PAN (p.14, lines 16-23) suggest you *are* discussing Fig. 5, but what I see is a strong vertical gradient at the tropopause, not a zone of mixed tropospheric and stratospheric characteristics. I read reference to a "small mixing layer around the tropopause" (p.14, line 33) which seems to minimize the significance of mixing here; if it's not significant (strong transport barrier), why spend a whole paragraph (lines 10-23) describing it?

There are some places where the English is a bit obscure to me, e.g. on
p.15, line 15, I don't know what "yield predominantly" means in this context. I wonder if the words "yield to" (e.g. on p.15, line 23) is intended to mean "serves to" or "results in", i.e. "serves to increase," or "results in increasing."

References:

Fairlie, T. D., M. A. Avery, R. B. Pierce, J. Al-Saadi, J. Dibb, and G. Sachse (2007), Impact of multiscale dynamical processes and mixing on the chemical composition of the upper troposphere and lower stratosphere during the Intercontinental Chemical Transport Experiment–North America, J. Geophys. Res., 112, D16S90, doi:10.1029/2006JD007923.

Garny, H., and W. J. Randel, Transport pathways from the Asian monsoon anticyclone to the stratosphere, Atmos. Chem. Phys., 16, 2703–2718, 2016, www.atmos-chem-phys.net/16/2703/2016/ doi:10.5194/acp-16-2703-2016

---

## Author Comment (AC1) · 2 Sep 2016

**Author Comment to Referee #1**

ACP Discussions doi: 10.5194/acp-2016-463
(Editor - Peter Haynes)
**'Long-range transport pathways of tropospheric source gases originating in Asia into the northern lower stratosphere during the Asian monsoon season 2012'**
* * *
We thank Referee #1 for further guidance on how to to revise our paper. Following the reviewers advice we have elaborated some minor points, which strengthen our findings. Our reply to the reviewer comments is listed in detail below. Questions and comments of the referee are shown in italics.

*This paper presents an analysis of the evolution and transport of air with emission sources in the Asian monsoon region to the northern hemisphere extratropical lower stratosphere using primarily output from the Chemical Lagrangian Model of the Stratosphere (CLaMS) driven with winds from the ERA-Interim reanalysis. Aircraft observations are used to establish confidence in the model simulations and demonstrate linkages between transport signatures and air masses observed in the extratopical lower stratosphere downstream. The main conclusions of the study are that the air lofted into the Asian monsoon anticyclone is confined to the tropical upper troposphere during the monsoon season, but is transported poleward into the extratropical lower stratosphere during the breakup of the Asian monsoon anticyclone in early fall via mostly Rossby wave breaking events over the Pacific and Atlantic ocean basins. The paper is well-detailed, well-structured, and well-reasoned. I do not find any fundamental errors in the analysis or questionable claims in the attendant discussion, but I believe the paper will benefit from a bit more cohesion of arguments throughout and a bit more analysis of the seasonality of transport. I recommend that the paper be considered for publication after mostly minor revisions. My general and specific comments to help guide the revisions are provided below.*

**General Comments**

*1. The linkages between transport of the monsoon upper troposphere air to the extratropical lower stratosphere and poleward Rossby wave breaking downstream of the anticyclone are clearly established in the manuscript. However, I feel the authors miss an opportunity to strengthen these points as their analysis expands in the latter part of the paper. Namely, the recent climatological studies of Rossby wave breaking cited in the paper provide support for the observed seasonality and timing of "flooding" of moist monsoon air into the extratropical lower stratosphere and for the dominant locations of poleward wave breaking events pointed out in Figure 16. Such additional detail on these linkages will provide better continuity in the Results section and strengthen the presentation and discussion of the conceptual model given in Figure 16.*

Many thanks for this suggestion. We agree that such details would strengthen our results and added therefore the following paragraph to the introduction:

"Exchange of air masses from the troposphere to the stratosphere occurs preferably in poleward flow structures such as tropospheric intrusions (e. g. Sprenger et al., 2007; Pan et al., 2009; Vogel et al., 2011). These intrusions develop into elongated potential vorticity (PV) streamers and are manifestation of Rossby wave breaking. Rossby wave breaking is identified as an important mechanism for exchange of air masses between the tropical upper troposphere and the extratropical lower stratosphere with a pronounced peak during summer in each hemisphere controlled by the presence of monsoon anticyclones, in particular of the Asian monsoon (e. g. Homeyer and Bowman, 2013; Kunz et al., 2015). High frequency of PV streamers are found over the eastern North Pacific and over the Atlantic in summer demonstrated in a recently published climatology of PV streamers (Kunz et al., 2015)."

*2. While the authors do a good job of including observational support for the model results with analysis of a few flights during the TACTS/ESMVal campaign, a more thorough test of the CLaMS model (particularly related to the influence of Asian monsoon air on extratropical lower stratosphere water vapor – i.e., Figure 15) using the same chemical dataset that was used for initialization – Aura MLS – is requested. Is the enhancement in extratropical lower stratosphere water vapor at 380 K following breakup of the Asian monsoon anticyclone an observed characteristic? I expect this is a*

*straight-forward test of the model and would go a long way in strengthening the perceived impact of this paper. If not an observed characteristic, this is a questionable result.*

We followed the reviewers advice and performed a comparison between CLaMS and MLS water vapor in the extra-tropical northern lower stratosphere. Overall, MLS water vapor measurements confirm our model results. Both MLS and CLaMS water vapor clearly show an increase of water vapor during summer and autumn 2012 in the northern extra-tropical lower stratosphere. However some differences between CLaMS and MLS remains. This differences are discusses within the revised version of our paper as follows.

"Because water vapor is an important greenhouse gas and even small perturbations of water vapor mixing ratios in the ExUTLS have a significant impact on surface climate, we are interested to estimate the impact of the Asian monsoon on moistening the lower stratosphere. From our simulations, we roughly estimate the fraction of $H_2O$ originating in India/China, Southeast Asia, and tropical Pacific Ocean contributing to the water budget in the lower northern hemisphere stratosphere. Fig. 1 shows the mean water vapor content in the northern lower stratosphere for PV values larger than $7.2\,PVU$ ($10\,PVU$) and northward of $30°\,N$ at $380\,K$ ($400\,K$) calculated with CLaMS (black line). In CLaMS at this altitude, contributions of cirrus clouds to the total water content are of minor importance. An increase of $H_2O$ in the northern lower stratosphere is found in our simulation during summer and autumn as reported in previous studies (e.g. Ploeger et al., 2013; Zahn et al., 2014; Müller et al., 2015). The fraction of $H_2O$ from different boundary tracers is indicated by different colors. End of October 2012, a contribution of approximately $1.5\,ppmv$ ($1.0\,ppmv$) $H_2O$ originates from source regions in Asia and the tropical Pacific compared to a mean water vapor content of $\approx$ $5\,ppm$ ($4.5\,ppm$) at $380\,K$ ($400\,K$). Total $H_2O$ without contributions of all boundary tracers i. e. the contribution of aged air (see Fig. 1, green line) shows a decrease during summer and autumn. Mean $H_2O$ from AURA-MLS (version 3.3 and version 4) in the northern stratosphere calculated similar as mean CLaMS $H_2O$ values for PV values larger than $7.2\,PVU$ ($10\,PVU$) and northward of $30°\,N$ at $380\,K$ ($400\,K$) also show an increase of water vapor within the lower northern stratosphere during summer and autumn 2012 (see Fig. 1, gray and purple line) and therefore support our findings from CLaMS simulations. Differences between CLaMS and MLS mean values in

the lower northern stratosphere in particular at $380\,\mathrm{K}$ could by explained by sampling issues (different spacial resolution of MLS and CLaMS), vertical resolution of MLS limiting measurements of steep tracer gradient around the tropopause and the initialization of CLaMS $H_2O$ at 1 May 2012 (above $400\,\mathrm{K}$: AURA-MLS; below $350\,\mathrm{K}$: CLaMS multi-annual simulation based on ERA-Interim water vapor with a linear transition between $350\,\mathrm{K}$ and $400\,\mathrm{K}$, more details see (Vogel et al., 2015)). Previous comparisons between CLaMS and MLS (v3.3) water vapor demonstrate that differences in $H_2O$ found in the lower stratosphere at high latitudes are likely an artifact of the MLS averaging kernels (Ploeger et al., 2013). Further, a comparison of water vapor climatologies from international limb sounder by Hegglin et al. (2013) demonstrate that Aura-MLS (v3.3) $H_2O$ measurements tend to be low at high latitudes in the lowermost stratosphere. Because of the good agreement between CLaMS $H_2O$ and in situ water vapor measurements by the FISH instrument during TACTS/ESMVal in the lower northern stratosphere (see Figs. 9-11) we are confident that CLaMS water vapor simulations within the extra-tropical lower stratosphere are reliable."

*3. The naming convention for emission sources outside of the Asian monsoon region is inconsistent in the text and Figures of the manuscript. While the use of "residual" is common and seems to be the primary intention of the authors, "residue" appears in other places (e.g., Figures 5, 6, 14, and 15; Table 2; and Page 12, line 23; Page 12, line 29). Please update the text and figures to refer to this as "residual" throughout.*

We agree that within the paper the naming convention should be consistent and would like to thank the reviewer for carefully reading the manuscript. We changed "residue" to "residual" overall in the revised version of the paper (both within text and figures).

*4. While the figures are (for the most part) visually appealing, an effort should be made to have the spacing, scaling, and text sizes consistent throughout. For example, the color bar labels run into the latitude axes in Figures 3 and 4, the cross-sections are misaligned in Figures 5 and 6, text sizes of the two panels in Figure 7 are different, the bottom panel is unnecessarily displaced from the top three in Figure 12, and the text sizes in Figures 14 and 15 are not legible at normal zoom.*

[Figure]

Figure 1: **(see Fig. 15 in ACPD paper)** The increase of $H_2O$ mixing ratios in the lower northern hemisphere stratosphere at $380\,K$ (top) and at $400\,K$ (bottom) during summer 2012 is shown (black line). The green line indicated $H_2O$ mixing ratios without fractions from the Earth's boundary layer. Mean $H_2O$ mixing ratios derived from MLS version v3.3 (gray) and version v4 (purple) in the northern extra-tropical lower stratosphere show also an increase in water vapor during summer and autumn. A rough estimation of the fraction of $H_2O$ mixing ratios originating in India/China (red), Southeast Asia (yellow) and the tropical Pacific Ocean (blue) in the northern lower stratosphere is also given.

We revised the figures according the reviewers advice.

**Specific Comments**

1. *Page 1, line 7: "...jet such as the..." should be "jet such that the"*
   done

2. *Page 1, lines 16-17: This statement is confusing here. This should be clarified to say that sources from Asia and the tropical Pacific account for ∼1.5 ppmv of the ∼5 ppmv mean in the extratropical lower stratosphere.*

   We revised the sentence as follows: "End of October 2012, approximately $1.5\,$ppmv $H_2O$ is found in the lower northern hemisphere stratosphere (at $380\,$K) from source regions both in Asia and in the tropical Pacific compared to a mean water vapor content of $\approx 5\,$ppmv."

3. *Page 2, line 3: "...is acting..." should be "...acts..."*
   done

4. *Page 3, lines 16-23: This discussion is limited to large-scale transport processes, correct? For example, we know that moist convection (apart from a large organized system like a typhoon) is capable of transporting air across the tropopause but such small-scale processes (though possibly represented to an extent) are not resolved in these models. A bit more detail and context should be given to clarify these points here, which you do reflect on near the end of the paper.*

   The following sentence is added: "In addition to this large-scale transport process of water vapor by air mass exchange between the tropics and the extratropics, convection induced injections of water vapor in mid-latitudes can also occur in large storm systems such as tropical cyclones and by deep continental convection (e. g. Anderson et al., 2012; Homeyer et al., 2014; Vogel et al., 2014)."

5. *Page 3, line 27: "...are used..." should be "...is used..."*
   done

6. *Page 4, line 14: "Measurements of..." should be "Measurements from..."*
   done

7. *Page 6, line 30: "...occurs like for all..." should be "...occurs in an equivalent fashion to all..."*

   on page 5, line 30: "...occurs in an equivalent way to all chemical species" is added

8. *Page 7, line 20: "...as for..." should be "...to..."*
   done

9. *Page 8, line 12: "...into stratosphere." should be "...into the stratosphere."*
   done

10. *Page 8, lines 15-18: The Homeyer et al 2011 paper you cite can be referenced here as well.*
    done

11. *Page 10, lines 11-12: What do you mean by this statement? It takes 5 weeks for the parcels to be transported from their surface emission locations to the lower stratosphere over Europe? This statement needs to be clarified a bit.*

    The following sentence is added: "These air masses uplifted by typhoon Bolaven are transported from the Earth's surface over the West Pacific within 5 weeks to the lower stratosphere over Europe [Vogel et al. 2014]."

12. *Page 10, lines 19-21: Based on the time series, No. 3 is an aircraft-only signature (i.e., no apparent plume in the CLaMS simulation – at least not to me!).*

   That is correct, therefore we revise the text as follows: "During the second part of the flight on 26 September 2012 (see Fig. 9, top), further signatures of tropospheric air are measured during the flight (No. 3-6). Also here, enhanced percentages of the emission tracers for India/China and Southeast Asia / tropical Pacific Ocean up to 18 % are simulated, except for region No. 3."

13. *Page 10, lines 26-18: But is this really dynamics of the AMA or of a downstream RWB event? You have already demonstrated that the latter is the reason this particular air mass crossed the tropopause, correct?*

   We revise the sentence (page 10, line 26-28) as follows: "Our simulations in agreement with measurements show that the amount of water vapor and pollution in the lower stratosphere is enhanced in the Northern Hemisphere in September 2012 associated with both the dynamic of the Asian monsoon anticyclone and the transport of air masses from both Asia and the tropical Pacific along the subtropical jet."

14. *Page 10, line 35: "Northern Hemisphere" should be "stratosphere"*
   revised to "... into the northern extratropical lower stratosphere..."

15. *Page 11, line 22: Remove "these"*
   done

16. *Page 12, line 25: "...masses on the..." should be "...masses to the...", and "End October..." should be "End of October..."*
   done

17. *Page 12, line 27: "Here, highest contributions are from tropical..." should be "Here, the highest contributions are from the tropical..."*
   done

18. *Page 12, line 31: "This is in particular..." should be "This is particularly..."*
   done

**References**

[revised manuscript text omitted]

---

## Author Comment (AC2) · 2 Sep 2016

**Author Comment to Referee #2**

ACP Discussions doi: 10.5194/acp-2016-463
(Editor - Peter Haynes)

'Long-range transport pathways of tropospheric source gases originating in Asia into the northern lower stratosphere during the Asian monsoon season 2012'
* * *
We thank Referee #2 for the very helpful review. Our reply to the reviewer comments is listed in detail below. Questions and comments of the referee are shown in italics.

*Summary: In this study the authors examine the transport characteristics associated with the Asian summer monsoon during September-October 2012, using both measurements of trace gases (e.g. ozone, water vapor, methane) and idealized tracer simulations that provide information about the relative contributions of different boundary layer regions to upper tropospheric/lower stratospheric air masses. The study provides strong evidence that the eastward shedding of eddies from the monsoon anticyclone provides an important mechanism for transporting boundary layer air from India and South Asia to the lower stratosphere over northern midlatitudes. While this study provides a comprehensive analysis and important contribution to the field that will make it suitable for publication, I have a few major comments that need addressing before I recommend its publication. In particular I have one major concern about the authors' interpretation of the air-mass origin tracers that needs addressing, as it may potentially affect the interpretation of the main results. I also have smaller comments that are, by comparison, less important.*

We thank the reviewer for the opportunity to remove unclarities about the emission tracer approach used in our paper. A detailed discussion follows below.

**Major Comment**

*1. I am concerned about the interpretation of the air-mass tracers as a fraction. It is definitely constructive to look at the relative contributions of*

*different source regions and I commend the authors' use of the diagnostic. However, more care should be taken in the interpretation of the tracer concentration as giving the fraction of air that was last at the earth's surface in a given source region. In particular, the simulation only covers 1 May 2012 - 31 October 2012. If the tracers are to be interpreted as fractions (as the authors intend them to be) then the sum of the air-mass tracers corresponding to all of the source regions must equal 1 (since the union of the source regions is the entire planetary boundary layer (PBL)). This is not the case, however, as shown in Figure 9. The sum of the red, blue and orange lines should, in principle, equal 1 (but does not). What this tells me is that the tracers have not been integrated to equilibrium so that there is a large amount of air that is not accounted for by the source fractions. This is a known issue when dealing with air-mass origin tracers (Orbe et al. (2015)) and I am concerned about what this means for the main conclusions in the study. Please either start the simulation much earlier (to ensure tracer equilibration by September 2012) or remove all references to "fraction" because this interpretation is not correct. Alternatively, it is possible that I am missing something important in the authors' definition of "residual" (by which I interpret the rest of the PBL) - if this is the case, please clarify in the text.*

*Orbe, Clara, Paul A. Newman, Darryn W. Waugh, Mark Holzer, Luke D. Oman, Feng Li, and Lorenzo M. Polvani. "Airmass Origin in the Arctic. Part I: Seasonality." Journal of Climate 28, no. 12 (2015): 4997-5014. (Figure 3b)*

The reviewers interpretation of the residual is correct, however we believe that there is a misunderstanding regarding an important point in our approach: The composition of air in the UTLS is always a combination of aged air masses from the stratosphere and free troposphere and young freshly injected air masses from the boundary layer. Only for the young air masses, our characterization of air mass origin is conducted.

In our approach we cannot run the atmosphere to equilibrium as we are interested in working out the contribution of young, freshly injected air to the composition of both the Asian monsoon anticyclone and the extratropical lower stratosphere. In so far our approach is different but complementary to the approach by Orbe et al..

We are aware that we had not brought this point across very well in our original paper (ACPD Version). We have now improved the presentation of

our modeling concept and its impact on the results throughout the paper (see in particular Section 3 in the revised version).

To go in more detail, the idea of using artificial tracers to identify transport has been used for a long time in both Eulerian (e.g. Mahlman and Moxim, 1978; Stone et al., 1999) and Lagrangian models (e.g. Bowman and Cohen, 1997; Bowman and Carrie, 2002; Günther et al., 2008; Vogel et al., 2011). In our approach, the artificial emission tracers in CLaMS are designed to identify possible boundary source regions in Asia that could contribute to the composition of the Asian monsoon anticyclone and the extratropical lower stratosphere during the course of the monsoon season 2012 considering advection and mixing processes.

The CLaMS transport scheme consists of advection, which is the reversible part of transport (the trajectory), and mixing, the irreversible part of transport. At beginning of the CLaMS model simulation each air particle is marked by a boundary layer tracer ($\Omega$). That means, for example, that the boundary layer tracer ($\Omega$) for air parcels in the boundary layer (BL) is set equal to one and everywhere else (free troposphere and stratosphere) equal to zero. Mixing in CLaMS between an air parcel from the boundary layer ($\Omega = 1$) and an air parcel from the free troposphere or stratosphere ($\Omega = 0$) is implemented by insertion of a new particle. The boundary layer tracer ($\Omega$) of the new particle is then equal to 0.5. Successive mixing processes between boundary layer air ('young air masses') and air masses from the free troposphere or stratosphere ('aged air masses') during the course of the simulation yield a tracer distribution differing from the initial distribution ($\Omega = 1$ or $\Omega = 0$). In the CLaMS simulation used here, air masses in the model boundary layer are marked by boundary emission tracers every 24 h (the time step for mixing in CLaMS) during the simulation period from 1 May 2012 until end of October 2012. Thus in our simulations, the age spectrum is relatively simply constructed from the response to a single pulse during $(t_i, t_i + \Delta t_i)$ for times $t > t_i$ (see Holzer et al., 2009), with $t_i$ equal to 1 May 2012 and $\Delta t_i$ equal to 6 months. This pulse marks 'young air masses'. E.g. a value of $\Omega$ equal to 0.4 means that $40\,\%$ of the air parcel is younger than $\Delta t_i$ (= 6 months; 'young air masses') and $60\,\%$ are older than $\Delta t_i$ released from the model boundary layer before 1 May 2012 (emission time $t_e < t_i$; 'aged air masses'). We used this single pulse approach because the focus of the paper is to analyze the influence of fresh emissions from different boundary regions using the meteorological conditions of the year 2012. Here, the air-mass fraction $f(r, t \mid \Omega)$ is defined to be the fraction of air at a location $r$ and time $t$

(with $t_i \leq t \leq t_i + \Delta t_i$) that was emitted since $t_i$ within $\Delta t_i$ from the model boundary layer (BL).

To get the information about the origin of young air masses within the model boundary layer, we divided the boundary layer (BL) in different regions, i.e. different artificial tracers of air mass origin, referred to as "emission tracers" ($\Omega_i$ of the number n = 17) are introduced in CLaMS (see paper table 1). Within the boundary layer, the sum of all different emission tracers ($\Omega_i$) is equal to 1 ($\Omega = \sum_{i=1}^{n} \Omega_i = 1$). During the course of the simulation, the air-mass fraction $f_i(r, t \mid \Omega_i)$ is defined to be the fraction of air at a location $r$ and time $t$ (with $t_i \leq t \leq t_i + \Delta t_i$) that was emitted since $t_i$ within $\Delta t_i$ from the model boundary layer (BL) in region $\Omega_i$. Note however that emission tracers only describe the contribution of the young air masses.

The fraction of air from different boundary regions $\sum_{i=1}^{n} f_i(r, t \mid \Omega_i) = f(r, t \mid \Omega)$ shown in Fig. 9 and also Figs. 10 and 11 (red, blue, orange) is less then 100 % because only "young air masses" emitted since 1 May 2012 are counted. Air masses originating in the free troposphere or stratosphere ('aged air masses') also contribute to the composition of the extratropical lower stratosphere. Thus in Fig. 9 region No.2 (gray area), the sum of air masses originating in India/China, in Southeast Asia/tropical Pacific and in the residual surface ($\sum_{i=1}^{n} f_i(r, t \mid \Omega_i)$) is roughly 45 %. The remaining 55 % of the composition of the lower stratosphere in this region is from the free troposphere and the stratosphere. Fig. 14 illustrate how the contribution of the model boundary layer ($f(r, t \mid \Omega)$) to the extratropical stratosphere rises to 44 % at 360 K, to 35 % at 380 K, and to 23 % at 400 K (see Sect. 4.4.2) from May until the End of October 2012. The remaining percentages to 100 % are contributions from "aged air masses" originating in the free troposphere and stratosphere.

We added the following sentence: 'On 26 September 2012 (see Fig. 9, top), a very pronounced signature of tropospheric air in the lower stratosphere is found between 09:05 UTC to 10:17 UTC (No. 2). Here, the contributions of the emission tracer for India/China and Southeast Asia /tropical Pacific Ocean are up to 20 % and 23 %, respectively (up to 5 % from the residual surface). Thus, the sum of all emission tracers for model boundary layer is roughly 48 %. The remaining 52 % of the composition of the lower stratosphere in this region is from aged air masses originating in the free troposphere and the stratosphere at the beginning our the CLaMS simulation on 1 May 2012.'

The technique used in our paper is different compared to the approach used in Orbe et al. (2015). A detailed comparison of our approach and the approach used in Orbe et al. (2015) is published in (Vogel et al., 2015, Sect. 4 Discussion). For clarification we repeat the corresponding paragraphs published in Vogel et al. (2015) here in our authors' comment:

"In contrast to our study, Orbe et al. (2015) used tracers of air mass origin in model simulations to infer the impact of boundary regions in Asia to the tropical lower stratosphere with an approach that the different tracers have equilibrated so that the sum of all tracers of air mass origin is equal to unity within the entire atmosphere (using a spin-up of 20 years). This approach provides for each air parcel the information about the origin within the boundary layer. However, this approach provides no information on the transport times from the boundary layer in Asia to the tropical lower stratosphere. To infer the transport times in the approach by Orbe et al. (2015), they introduce a boundary impulse response that marks air that left the boundary layer on a certain day (1 July). They infer transport times of 1 month from the boundary layer in Asia into the tropical lower stratosphere in July when the Asian monsoon is active. In this respect, their pulse serves a similar purpose as our seasonal tracer set-up. However, our approach considers all air masses that left the boundary layer over the course of the 2012 monsoon season since May and thus reflects the meteorological conditions of the entire 2012 monsoon season."

**Specific Comments**

1. *Line 29, Page 3: I am wary of the use of the term "transport pathways." The air mass origin tracers only tell you where air was last in contact with the boundary layer. They do not provide a sense for how the air arrived at the receptor location, so please remove all references to pathways. To infer pathways you would need to use idealized tracers similar to those used for inferring the age spectrum or, most appropriately, the path density tracers examined in Holzer (2009)*

   *Holzer, Mark. "The path density of interhemispheric surface-to-surface transport. Part I: Development of the diagnostic and illustration with*

*an analytic model." Journal of the Atmospheric Sciences 66, no. 8 (2009): 2159-2171*

Our approach allows the relative contribution of air masses of varying origin to be quantified for every individual CLaMS air parcel on a certain location ($r_j$) and for each time ($t_l$) during the course of the simulation. Thus, transport pathways from the boundary layer into the UTLS (occurring since 1 May 2012) can be inferred analyzing the spatial-temporal evolution of artificial emission tracers during the simulation period. The change of the fraction of air originating in region $\Omega_i$ on different locations ($r_j$) for different times ($t_l$) $f_{ijl}(r_l, t_j \mid \Omega_i)$ define in our approach the transport pathways. However, we agree with the reviewer that the term "pathway" might have been used a bit to excessively and have removed the word on various locations in the paper.

2. *Line 8, page 6: Again, reservation about the word "pathway."*

okay, we dropped "pathway"

3. *Lines 33-35, page 6: The sum of all of the air-mass fractions does not equal 1, leaving a large fraction of air unaccounted for. Therefore, I am not confident in the statement that "that air masses originating in India/China and Southeast Asia/Pacific Ocean almost exclusively contribute to the chemical composition of the separated anticyclone." Please either start your simulation earlier (to ensure equilibration of the airmass tracers) or do not use the word "fraction".*

We agree the air masses originating in India/China and Southeast Asia/Pacific Ocean do not exclusively contribute to the chemical composition of the separated anticyclone. For clarification we added the following sentence to the paper:
"Further, the emission tracer for Southeast Asia / tropical Pacific Ocean contributes to the composition of the Asian monsoon anticyclone, however to a smaller extent compared to the emission tracer for India/China as shown in Fig. 4 (right). Contributions from the residual surface are of minor importance (see Sect. 4.2). Note that contribution of air masses originating in the free troposphere and stratosphere

('aged air masses') als contribute to the composition of the Asian monsoon anticyclone of about 25 % at the end of September 2012 (see in Vogel et al., 2015, Fig. 9)."

**References**

Bowman, K. P. and Carrie, G. D.: The mean-meridional transport circulation of the troposphere in an idealized GCM, J. Atmos. Sci., 59, 2002.

Bowman, K. P. and Cohen, P. J.: Interhemispheric exchange by seasonal modulation of the Hadley circulation, J. Atmos. Sci., 54, 1997.

Günther, G., Müller, R., von Hobe, M., Stroh, F., Konopka, P., and Volk, C. M.: Quantification of transport across the boundary of the lower stratospheric vortex during Arctic winter 2002/2003, Atmos. Chem. Phys., 8, 3655–3670, URL http://www.atmos-chem-phys.net/8/3655/2008/, 2008.

Mahlman, J. and Moxim, W.: Tracer simulation using a global general circulation model - results from a mid-latitude instantaneous source experiment, J. Atmos. Sci., 35, 1340–1374, 1978.

Orbe, C., Waugh, D. W., and Newman, P. A.: Air-mass origin in the tropical lower stratosphere: The influence of Asian boundary layer air, Geophys. Res. Lett., 42, 2015GL063937, doi:10.1002/2015GL063937, URL http://dx.doi.org/10.1002/2015GL063937, 2015.

Stone, E. M., Randel, W. J., and l. Stanford, J.: Transport of passive tracers in baroclinic wave life cycles, J. Atmos. Sci., 56, 1364–1381, 1999.

Vogel, B., Pan, L. L., Konopka, P., Günther, G., Müller, R., Hall, W., Campos, T., Pollack, I., Weinheimer, A., Wei, J., Atlas, E. L., and Bowman, K. P.: Transport pathways and signatures of mixing in the extratropical tropopause region derived from Lagrangian model simulations, J. Geophys. Res., 116, D05306, doi:10.1029/2010JD014876, 2011.

Vogel, B., Günther, G., Müller, R., Grooß, J.-U., and Riese, M.: Impact of different Asian source regions on the composition of the Asian monsoon anticyclone and of the extratropical lowermost stratosphere, Atmos. Chem.

Phys., 15, 13 699–13 716, doi:10.5194/acp-15-13699-2015, URL `http://www.atmos-chem-phys.net/15/13699/2015/`, 2015.

---

## Author Comment (AC3) · 6 Sep 2016

**Author Comment to Referee #3**

ACP Discussions doi: 10.5194/acp-2016-463
(Editor - Peter Haynes)
**'Long-range transport pathways of tropospheric source gases originating in Asia into the northern lower stratosphere during the Asian monsoon season 2012'**
* * *
We thank Referee #3 for the good evaluation of our paper. Following the reviewers advice we revised some parts of the paper for the purpose of clarification. Our reply to the reviewer comments is listed in detail below. Questions and comments of the referee are shown in italics.

*The authors use the global Lagrangian CLaMS model, with artificial and chemical constituent tracers to quantify the contributions of different boundary Layer (BL) source regions in Asia to the Asian Summer Monsoon (ASM) anticyclone, and from there to extra-tropical lower stratosphere (ExLS), for the 2012 ASM season. Further, they illustrate the transport pathways for BL source air, accumulated in the ASM anticyclone, to reach the ExLS, via eddy shedding in the upper troposphere, subsequent filamentation and penetration into the stratosphere, associated with Rossby wave breaking along the subtropical jet. They also consider the westward shedding of air from the ASM anticyclone into the tropical upper troposphere.*

*The authors use the simulated artificial tracers, and simulated ozone, CO, and water vapor, to interpret small-scale structures observed along aircraft flight tracks as filamentary intrusions of BL source air associated with the ASM anticyclone into the lower stratosphere over Northern Europe. Further, they use the artificial tracers to quantify the contribution of different BL source regions to the ExLS over the 2012 ASM season, and use CLaMS simulated water vapor to estimate the contribution of Asian source regions to water vapor in the ExLS.*

*The study builds upon earlier studies that have illustrated troposphere-stratosphere exchange (STE) mechanisms associated with the ASM, going*

*back at least as far as Dethof et al. (1999), a paper which the authors cite. At the same time the use of the artificial and constituent tracers with the CLaMS model to quantify estimates of the contributions from Asian (and other) BL source regions to the ExLS is I believe a step beyond these earlier studies. Quantification of water vapor contribution to the NH lower stratosphere (p.13) is particularly interesting. The cross-section of the filamentary structure (Fig. 6) provides an illuminating illustration of intermediate (mixed) constituent and stability conditions between the tropospheric and stratospheric air mass characteristics.*

*The comparison of aircraft observations of the low stratosphere over Northern Europe with CLaMS tracer maps interpolated to the aircraft flight tracks illustrates effectively that the simulation of tropospheric filaments in the low stratosphere represents real-world conditions, and supports the quantitative estimates of BL source influence in the ExLS presented later. The analysis and interpretation is reminiscent of that conducted by Fairlie et al. (2007) for INTEX-NA aircraft observations; the authors may wish to add correlation scatter plots of the observed O3, CO or CH4, water vapor to further illuminate air mass origin and characteristics of mixed troposphere-stratosphere air masses.*

*I think the paper could use an editorial review for the English and sentence structure. There is occasional awkwardness in the sentence structure, and some choice of wording that I find confusing, and may be a translation issue (see some examples below). Nevertheless, I think the paper is suitable for publication in ACP given consideration to these issues and the points listed below, most of which are minor and for the purpose of clarification.*

We agree with the reviewer that tracer-tracer correlations are a powerful technique to study mixing processes between the troposphere and stratosphere. Tracer-tracer correlations would integrate the information from the aircraft observations and the CLaMS model to quantify and characterize signatures of mixing between stratospheric and tropospheric air (e.g. monsoon air) and therefore give insights in single mixing processes. However, in this paper we didn't show tracer-correlations because that would be material for an additional new study that we are planing. Therefore, to analyze tracer-tracer correlations of different chemical species from different TACTS/ESMVal measurements and tracer-tracer correlations from the CLaMS model using in

addition tracers of air mass origin (e.g. similar as in Vogel et al. (2011)) would go beyond the scope of the paper discussed here. For the TACTS flight on 30 August 2012 an analysis of tracer-tracer correlations ($O_3$- CO) is already published in Müller et al. (2016).

We would like to thank the reviewer for carefully reading our paper and identifying some parts of the paper that need revisions for the purpose of clarification (details see below 'Minor Comments'). For the most comments identifying ambiguities, the reviewers interpretation was correct (see below). Following the reviewers advice, we revise those parts of the text. In general, some English corrections are already made in the revised version of the paper. After the acceptance of our paper by ACP a language revision of the paper will be made by the Language Services of Forschungszentrum Jülich.

**Minor Comments**

1. *p.6, line 31, What is meant by "Maritime Continent"?*

   The expression "The Maritime Continent" describes the region between the Indian and Pacific Oceans including the archipelagos of Indonesia, Borneo, New Guinea, the Philippine Islands, the Malay Peninsula, and the surrounding seas.

   We added in the revised version of the paper: '...Maritime Continent (the region between Indian and Pacific Oceans)...'

2. *p.7, line 12-13: Comment: The authors will recognize that the transport is only irreversible if the tropospheric intrusion is mixed into the stratospheric surroundings. It is conceivable that an intrusion across the PV=7.2 PVU could return to the troposphere downstream.*

   We agree that the sentence can be misinterpreted, therefore we remove 'irreversible' and write in the revised version: 'Isentropic transport of air masses across the 7.2 PVU isoline indicates exchange between the tropics and extratropics due to wave breaking.'

3. *p.7, lines 14-15: Comment: I am unable to see the PV=7.2 PVU isopleth enveloping a "region of enhanced tracers"*

We agree that the sentence is somewhat unclear. We revised the sentence as follows: 'On 20 September 2012, at the northern flank of the separated anticyclone the 7.2 PVU isoline is in the region where strongest gradients of the emission tracers occur indicating the transport barrier at $380\,\mathrm{K}$ (see Fig. 3 and 4).'

4. *p.7, line 31, instead of "surface that is" do you mean "surface, i.e.,"? I.e., are the authors stating the definition of "residual" here?*

For clarification we introduced a definition for the 'residual surface' in Sect. 3 (CLaMS simulations using artificial tracers of air mass origin) as follows:

'The most important regions for our study are India/China (= Northern India (NIN) + Southern India (SIN) + Eastern China (ECH)), Southeast Asia (SEA) and the tropical Pacific Ocean (TPO). The sum of all model boundary layer tracers ($\Omega = \sum_{i=1}^{n} \Omega_i$) without contributions from India/China, Southeast Asia, and the tropical Pacific Ocean is summarized in one emission tracer referred to as "the residual surface" (= $\Omega$ - NIN - SIN - ECH - SEA - TPO).'

and revised the sentence on page 7, line 31, Sect. 4.1, as follows:

'Fig. 5 (bottom left) shows the emission tracer for the residual surface (the entire model boundary layer without contributions from India/China, Southeast Asia, and the tropical Pacific Ocean, see Sect. 3).'

5. *p.7, line 32-33, reference "no signature." Would the authors be more quantitative here? Looks like up to 10-15% is due to "residual" sources in the anticyclone.*

We revised the sentence as follows: 'The contribution of the emission tracer for the residual surface is approximately 10-15 % within the separated anticyclone indicating that air masses originating in India/China

and Southeast Asia / tropical Pacific Ocean almost exclusively contribute to the chemical composition of the separated anticyclone.'

6. *p.8, line 12, reference "indicating transport from the troposphere into the stratosphere." This is according to the definition of the authors, based on the work of Kunz et al.*

Yes, that is correct. Therefore we revised the sentence to be more precise as follows: 'At 380 K enhanced contributions of the emission tracer from India/China are also found north of the 7.2 PVU barrier indicating transport from the troposphere into the stratosphere according to the definition by Kunz et al. (2015).'

7. *p.9, reference discussion of Fig. 9 emission tracer plots, here and elsewhere. Please confirm for the reader if "residual" includes all BL surfaces other than those identified (China/India, SEAsia/ tropical Pacific). How should the reader interpret the sum of these percentages being much less than 100%, e.g. does the remainder comprise background lower stratospheric air, unconnected to any BL surface in past 5 months?*

As mentioned above we added a definition for 'residual surface' in Sect. 3 (CLaMS simulations using artificial tracers of air mass origin) in the revised version of the paper. The interpretation that the sum of the percentages of all boundary layer tracers is less than 100% because of contributions of aged air masses from the stratosphere is correct. For further clarification we added the following sentence in the revised version:

'On 26 September 2012 (see Fig. 9, top), a very pronounced signature of tropospheric air in the lower stratosphere is found between 09:05 UTC to 10:17 UTC (No. 2). Here, the contributions of the emission tracer for India/China and Southeast Asia /tropical Pacific Ocean are up to 20 % and 23 %, respectively (up to 5 % from the residual surface). Thus, the sum of all emission tracers for model boundary layer is roughly 48 %. The remaining 52 % of the composition of the lower stratosphere in this region is from aged air masses originating in the free troposphere and the stratosphere at the beginning our the CLaMS simulation on 1 May

2012.'

8. *p. 10, discussion of Figs. 9-11. It would be helpful if the authors labeled the locations of flight segments "1", "2", "3", etc. on the maps in Figs. 7-8, to help the reader identify the features highlighted in the flight data to features on the CLAMS maps. Additionally, the flight data appears to be higher temporal resolution than the CLAMS profiles (e.g. the profile of FISH H2O). It may be helpful to add a time-averaged data profile at the same resolution as the CLaMS for better comparison. Tracer-tracer correlation plots may also be a useful addition (see above).*

We agree that numbers in Fig. 7-8 can help the reader to identify the regions that are impacted by young tropospheric air masses. We revised figures 7-8 and also Fig. 10 and 11 as shown below.

Within Fig. 9-11, the measurements of CO, $CH_4$, $H_2O$ and $O_3$ are shown in the highest resolution that was available for each species to demonstrate the variability of the measurements. For the interpolation of CLaMS results as described in Appendix B, backward trajectories are calculated every second along the flight path.

As discussed above an analysis of tracer-tracer correlations would be beyond the scope of this paper.

9. *p. 11, lines 13-15. Suggestion: I think the authors mean to emphasize the locations (Atlantic and Pacific Oceans) here, rather than the mechanism (Rossby wave breaking). They may want to leave out "Rossby wave breaking" in this sentence to keep the stress on the locations.*

That is a good point. We changed the sentence as follow: 'Finally, Fig. 11 shows that transport from air masses originating in the Asian monsoon anticyclone into the lower stratosphere take place most frequently over the Pacific and Atlantic Ocean ....'

10. *p.11, lines 18-20. This sentence seems a bit out of context here. Perhaps reference to SE Asia/ Tropical Pacific contribution (the appendix) would sit better after the introduction to Fig.12 (p.11, line 2, after "September 2012").*

[Figure]

Figure 1: (= **Fig. 7 within the ACPD paper**) Horizontal (top) cross-section of the fraction of air originating in India/China over Europe om 26 September 2012. The flight path transferred to noontime of the TACTS/ESMVal flight is shown as yellow line. Segments of the flight in the lower stratosphere with enhanced measured CO, $CH_4$, and $H_2O$ and reduced $O_3$ compared to the stratospheric background are highlighted in red and numbered for clarification (the same numbers are used in Fig. 9 The climatological isentropic transport barrier of 7.2 PVU at 380 K is shown as thick black line. The white thick line marks the position of the vertical (bottom) cross-section at 8°W longitude which is similar to the cross-sections shown in Fig. 5 and 6.

We agree. The paragraph is shifted as proposed.

11. *p.11, reference discussion of "transport pathways": The title of 4.4.1 is "transport pathways into the lower stratosphere." The authors have illustrated that "eastward eddy shedding" on the NE side of the anticyclone, and subsequent transport and filamentation of material can be a pathway to reach the stratosphere (pathway 1, line 10). But, what about the "westward eddy shedding" (pathway 2, line 11) from the anticyclone to the TTL? I find no discussion of this as a potential pathway to the stratosphere, via e.g. diabatic ascent (Garny and Randel, 2015).*

We agree that pathway 2 needs some further discussion and added therefore the following paragraph: 'Transport of air masses from the Asian monsoon anticyclone southeastwards into the tropics by e. g. westward eddy shedding causes increase of contributions of air masses originating in the boundary layer in India/China within the TTL. These air masses could penetrate into the upwelling in the deep branch of the Brewer-Dobson circulation and thus could be transported further up into the stratosphere (e. g. Garny and Randel, 2016).'

12. *p.12, discussion of Fig. 14 and Table 2. Please clarify how these metrics are computed. Are they achieved by area weighting daily isentropic "fraction of air" maps for areas north of 30 o N and for PV greater than the "transport barrier" PV? Are you saying for example that by end October 2012, almost 20% of the air in the NH at 360K north of these delimiters originates in the India/China BL within the previous 5 months?*

The reviewers interpretation of the percentages in Fig. 14. is correct. For clarification we revised the text as follows:

'The accumulation of young air masses from Asia since 1 May 2012 in the extratropical lower stratosphere is calculated using the isentropic transport barrier at different levels of potential temperature derived by Kunz et al. (2015) as shown in Fig. 14 and Tab. 2. To calculate the percentages of different emission tracers within the lower extratropical stratosphere at a certain level of potential temperature, we use the following approach: For each day between 1 May and 31 October 2012 a

mean value for each emission tracers of all CLaMS air parcels is calculated for PV values larger (lower) than those at the transport barrier (see Tab. 2) and for air masses poleward of 30° N (30° S) at a specific isentropic level ($\Theta \pm$ 0.5 K). End of October 2012, the contributions of all boundary emission tracers on the composition of the extratropical northern lower stratosphere are at 360 K $\approx$44%, at 380 K $\approx$35%, and at 400 K $\approx$ 23%, with the remaining fraction of air consisting of aged air. The highest contributions of the boundary emission tracers are from India/China uplifted within the Asian monsoon anticyclone, from Southeast Asia, and from the tropical Pacific Ocean. The contribution of all other regions of the Earth's surface (residual surface) are of minor importance (Tab. 2).'

13. *p.14, lines 12-14, reference "A mixing layer .... (see Fig. 5)" It is unclear to me what feature is being identified in Fig. 5. I see no discussion of such a mixing layer in earlier discussion of Fig. 5. Indeed 2 lines earlier (p.14, line 11) the thermal tropopause is described as a "strong transport barrier above the separated anticyclone." Do you really mean Fig. 5? Are you referring to the thin layer of strong vertical gradients in fractions of air and in simulated CO at the thermal tropopause south of 40 o N, 370-400K)? What is the evidence for mixing, and what is the mechanism? Or do you mean Fig. 6 instead where mixed troposphere-stratosphere characteristics of BL source fractions, CO, and buoyancy frequency are evident between the double thermal tropopauses? The discussion of PAN (p.14, lines 16-23) suggest you are discussing Fig. 5, but what I see is a strong vertical gradient at the tropopause, not a zone of mixed tropospheric and stratospheric characteristics. I read reference to a "small mixing layer around the tropopause" (p.14, line 33) which seems to minimize the significance of mixing here; if it's not significant (strong transport barrier), why spend a whole paragraph (lines 10-23) describing it?*

We agree the discussion about the mixing layer is redundant. Therefore, we remove this paragraph within the discussion section as well as the statement "small mixing layer around the tropopause" (p.14, line 33) within the conclusions.

14. *There are some places where the English is a bit obscure to me, e.g.*

*on p.15, line 15, I don't know what "yield predominantly" means in this context. I wonder if the words "yield to" (e.g. on p.15, line 23) is intended to mean "serves to" or "results in", i.e. "serves to increase," or "results in increasing."*

The sentence is revised as follows: 'This second transport pathway is mainly caused by westward eddy shedding, however this transport pathway enhances predominantly the contribution of fresh emission from India/China to the composition of TTL air at 380 K in summer and autumn 2012.'

**References**

Garny, H. and Randel, W. J.: Transport pathways from the Asian monsoon anticyclone to the stratosphere, Atmos. Chem. Phys., 16, 2703–2718, doi: 10.5194/acp-16-2703-2016, 2016.

Kunz, A., Sprenger, M., and Wernli, H.: Climatology of potential vorticity streamers and associated isentropic transport pathways across PV gradient barriers, J. Geophys. Res., 120, 3802–3821, doi:10.1002/2014JD022615, URL http://dx.doi.org/10.1002/2014JD022615, 2014JD022615, 2015.

Müller, S., Hoor, P., Bozem, H., Gute, E., Vogel, B., Zahn, A., Bönisch, H., Keber, T., Krämer, M., Rolf, C., Riese, M., Schlager, H., and Engel, A.: Impact of the Asian monsoon on the extratropical lower stratosphere: trace gas observations during TACTS over Europe 2012, Atmos. Chem. Phys., 16, 10 573–10 589, doi:10.5194/acp-16-10573-2016, 2016.

Vogel, B., Pan, L. L., Konopka, P., Günther, G., Müller, R., Hall, W., Campos, T., Pollack, I., Weinheimer, A., Wei, J., Atlas, E. L., and Bowman, K. P.: Transport pathways and signatures of mixing in the extratropical tropopause region derived from Lagrangian model simulations, J. Geophys. Res., 116, D05306, doi:10.1029/2010JD014876, 2011.

[Figure]

Figure 2: (= **Fig. 8 within the ACPD paper**) Horizontal cross-section of the fraction of air originating in India/China over Europe on 25 September at 370 K (top) and on 23 September 2012 at 380 K (bottom). The flight paths transferred to noontime for the TACTS/EsmVal flights 2012 over northern Europe are marked as yellow lines. Segments of the flight in the lower stratosphere with enhanced measured CO, $CH_4$, and $H_2O$ and reduced $O_3$ compared to the stratospheric background are highlighted in red and numbered for clarification (the same numbers are used in Fig. 10 and 11). The climatological isentropic transport barrier of 6.0 PVU (at 370 K) and 7.2 PVU (at 380 K) are shown as thick black lines.

[Figure]

Figure 3: (= **Fig. 10 within the ACPD paper**) As Fig. 9, but for the flight on 25 September 2012.

[Figure]

Figure 4: (= **Fig. 11 within the ACPD paper**) As Fig. 9, but for the flight on 23 September 2012.

---

## Author Response (AR2)

**Letter to the Editor**

ACP Discussions doi: 10.5194/acp-2016-463 (Editor - Peter Haynes) 'Long-range transport pathways of tropospheric source gases originating in Asia into the northern lower stratosphere during the Asian monsoon season 2012'

Dear Peter Haynes,

again many thanks for handling our manuscript. We prepared a revised version of our manuscript including all technical corrections proposed by Referee #1 (Report #2). A document specifying all changes in the final version compared to the last version is added.

Best wishes

Bärbel Vogel

**Long-range transport pathways of tropospheric source gases originating in Asia into the northern lower stratosphere during the Asian monsoon season 2012**

Bärbel Vogel1, Gebhard Günther1, Rolf Müller1, Jens-Uwe Grooβ1, Armin Afchine1, Heiko Bozem2, Peter Hoor2, Martina Krämer1, Stefan Müller2, Martin Riese1, Christian Rolf1, Nicole Spelten1, Gabriele P. Stiller3, Jörn Ungermann1, and Andreas Zahn3

[revised manuscript text omitted]
| Southern India (SIN)          | $0-20^{\circ}$ N                            | 55–90° E                       |
| Eastern China (ECH)           | $20-40^{\circ}$ N                           | 90–125° E                      |
| Southeast Asia (SEA)          | $12^{\circ} \text{ S}-20^{\circ} \text{ N}$ | 90–155° E                      |
| Siberia (SIB)                 | 40–75° N                                    | 55–180° E                      |
| Europe (EUR)                  | 45–75° N                                    | $20^{\circ}$ W–55 $^{\circ}$ E |
| Mediterranean (MED)           | $35-45^{\circ}$ N                           | $20^{\circ}$ W– $55^{\circ}$ E |
| Northern Africa (NAF)         | 0–35° N                                     | $20^{\circ}$ W–55 $^{\circ}$ E |
| Southern Africa (SAF)         | $36^{\circ} \text{ S-}0^{\circ} \text{ N}$  | 7–42° E                        |
| Madagascar (MDG)              | 27–12° S                                    | 42–52° E                       |
| Australia (AUS)               | 40–12° S                                    | 110–155° E                     |
| North America (NAM)           | 15–75° N                                    | $160-50^{\circ} \mathrm{W}$    |
| South America (SAM)           | $55^{\circ}$ S-15° N                        | 80–35° W                       |
| Tropical Pacific Ocean (TPO)  | $20^{\circ} \text{ S}-20^{\circ} \text{ N}$ | see Fig. 1                     |
| Tropical Atlantic Ocean (TAO) | $20^{\circ} \text{ S}-20^{\circ} \text{ N}$ | see Fig. 1                     |
| Tropical Indian Ocean (TIO)   | $20^{\circ} \text{ S}-20^{\circ} \text{ N}$ | see Fig. 1                     |
| Background                    | remaining surface                           |                                |